# Shape still matters: Rockfall interactions with trees and deadwood in a mountain forest uncover a new facet of rock shape dependency

Adrian Ringenbach[1,2,4], Peter Bebi[1,2], Perry Bartelt[1,2], Andreas Rigling[3,4], Marc Christen[1,2], Yves Bühler[1,2], Andreas Stoffel[1,2], and Andrin Caviezel[1,2]

[1]Climate Change, Extremes and Natural Hazards in Alpine Regions Research Center CERC, 7260 Davos Dorf, Switzerland
[2]WSL Institute for Snow and Avalanche Research SLF, 7260 Davos Dorf, Switzerland
[3]Swiss Federal Institute for Forest, Snow and Landscape Research WSL, 8903 Birmensdorf, Switzerland
[4]Forest Ecology, Institute of Terrestrial Ecosystems, Department of Environmental Systems Science, ETH Zurich, 8092 Zurich, Switzerland

**Correspondence:** Adrian Ringenbach (adrian.ringenbach@slf.ch)

**Abstract.** Mountain forests have a substantial protective function in preventing natural hazards, in part due to the presence of dead wood on the forest floor. Rates of deadwood accumulation have increased within the Alps and are predicted to rise further, due to natural disturbances. In particular, higher windthrow event frequencies are expected, primarily due to large-scale even-aged forest stands in many alpine regions combined with climate change. We quantified the rockfall protection effect of mountain forests with and without deadwood in unprecedented detail. in experiments using two rock shapes with important hazard potential and masses of 200-3200 kg. Based on a multi-camera setup, pre-and post-experimentally retrieved high-resolution lidar data, and rock data measured in situ, we precisely reconstructed 63 trajectories. The principal parameters of interest describing the rockfall kinematics were retrieved for each trajectory. A total of 164 tree impacts and 55 deadwood impacts were observed, and the currently applied energy absorption curves – partially only derived theoretically – could consequently be corroborated or even expanded to a greater absorption performance of certain species than hitherto assumed. Standing trees in general and deadwood, in particular, were found to strongly impede the notorious lateral spreading of platy rocks. Platy rocks featured a shorter mean run-out distance than their compact counterparts of similar weight, even in the absence of deadwood. These results indicate that the higher hazard potential of platy rocks compared with more compact rocks, previously postulated for open field terrain, applies less to forested areas. Lastly, reproducing the experimental setting showcases how complex forest states can be treated within rockfall simulations. Overall, the results of this study highlight the importance of incorporating horizontal forest structures accurately in simulations in order to obtain realistic deposition patterns.

## 1 Introduction

Forests with a protective function serve as cost-effective (Moos et al., 2017; Getzner et al., 2017; Olschewski et al., 2012) and widespread (Dupire et al., 2020; Brändli et al., 2020) nature-based protection infrastructure within alpine regions. The amount of deadwood in Swiss mountain forests increased during the last decades (Abegg et al., 2021). Additionally, natural disturbances in forests are expected to increase under future climate conditions (Seidl et al., 2017), leading to higher deadwood

accumulation rates. Therefore, it is of social and macroeconomic relevance to assess the influence of deadwood on the dynamics of natural hazards.

The important role of mountain forests as ecological infrastructure providing protection against rockfall is widely recognized. In New Zealand, for instance, researchers showed that deforestation resulting from human activities caused Anthropocene rockfalls to have longer run-out distances than prehistoric events (Borella et al., 2016). In several European studies, the maximum energy absorbed by living Norway spruce (Lundström et al., 2009; Kalberer et al., 2007) and silver fir (Dorren and Berger, 2005) has been described and used to derive three-dimensional rockfall models that include single trees absorbing the kinetic energy of consecutive impacts (Rammer et al., 2010; Dorren, 2012; Toe et al., 2017; Lu et al., 2020). Such models make it possible to simulate the protective effect of mountain forests on different scales – from a single slope to an entire region (Stoffel et al., 2006; Woltjer et al., 2008; Moos et al., 2017; Dupire et al., 2016; Lanfranconi et al., 2020). The influence of natural disturbances, such as windthrow, bark-beetle calamities, and forest fires, on the protective function of forests, has long been neglected in rockfall models. Their absence in numerical tools has often been due to an insufficient capacity to handle such three-dimensional structures from an input perspective, but equally to insufficient knowledge about their protective performance and effect. More recently, rockfall models have explicitly considered the influence of deadwood resulting from natural disturbances (Fuhr et al., 2015; Costa et al., 2021; Ringenbach et al., 2022c, a), resulting in shorter mean run out distances. In contrast to the integration of standing trees into rockfall models, however, incorporating lying deadwood has a limited experimental basis (Ringenbach et al., 2022c; Bourrier et al., 2012). The experimental rockfall data set presented here follows a systematic approach unprecedented for forests containing deadwood and, thus, contributes to closing the remaining research gap. The approach allows rockfall calibrations at different scales and is a vital link to real-world events.

Our experiment has been designed not only to elucidate the effect of deadwood on rockfall runout but also to consider other factors that may influence this metric. They include rock size and shape, and the moisture saturation of the slope materials along the runout path. Recent numerical work that focused on single rock–tree impacts reported an energy transfer from translational kinetic energy $E_{kin,trans}$ to the rotational kinetic energy $E_{kin,rot}$ for eccentric rocks but highlighted the need for experimental confirmation (Lu et al., 2020). Additionally, no systematic deposition pattern analysis with respect to rock size has been presented for forested slopes so far. Recently, the importance of such analyses was demonstrated by Caviezel et al. (2021a), who found that the downslope pathways of platy rocks have a larger lateral spread in open land compared to their mass-equivalent cubic-shaped counterparts. Combined with a comparable mean run-out distance and a similar release probability, this means that platy rocks pose an increased hazard potential in unobstructed terrain. Other rockfall experiments conducted within a quarry indicated that the berms acted as pronounced topographic features and were responsible for most of the deposition pattern. However, the shape of the rocks was the only attribute that could qualitatively influence the runout distance (Bourrier et al., 2021). Thus, scrutinizing the effect of rock shape on forested slopes and the compliance or deviation from the previously examined behavior in open land is another compelling motivation for conducting this experimental campaign.

The significance of the rock–substrate interaction has been recognized for open land in various rockfall simulations (Lu et al., 2019; Noël et al., 2021). A potentially important parameter, soil moisture, has received less attention so far. A study in New Zealand compared the experimentally derived rockfall run-out distances under known soil moisture conditions with those re-

ported during a nearby earthquake. The thereof-calibrated rockfall simulations imply that rockfalls under dry soil conditions achieve greater runout distances than rockfalls on the same soil when wet (Vick et al., 2019). This topic could become increasingly important under future climatic conditions with reduced precipitation if rockfall run-out distances are altered because of changing rock-soil interactions. In this paper, a first attempt is made to directly link rockfall kinematics with the different, on-site measured soil moisture. The newly acquired, detailed insights provide valuable context to the results obtained this far. In summary, we investigate the spatially explicit rockfall kinematics depending on the state of the forest including either lying deadwood or exclusively standing trees. Based on three mass classes, we resolve the rock shape effect within forests. Finally, the soil moisture is compared to the in situ measured accelerations and the reconstructed rock velocities.

## 2  Experimental test site and methods

### 2.1  Experimental site Schraubachtobel

Here, we present a systematic experimental rockfall campaign on forested slope for scenarios with lying deadwood (DW) and after the windthrow area was cleared (CLR). The experimental site Schraubachtobel (46.58248° N, 9.42303° E) is located roughly 20 km northeast of Chur, within the municipality of Schiers, Switzerland (Fig. 1a). The release point area is located at 860 m a.s.l., with the 245 m long and roughly 40 - 60 m wide experimental slope dropping over 164 m of elevation down to the riverbed of the creek *Schraubach* at 706 m a.s.l. The slope comprises steep terrain with an overall inclination along the central fall path of $\alpha = 39°$ and single slope sections of $\alpha_1 = 48°$, $\alpha_2 = 39°$, $\alpha_3 = 33°$, $\alpha_4 = 38°$ and $\alpha_5 = 43°$ (see their locations in Fig. 1b). The slope is covered by a regolith and soil layer with varying local thicknesses underlain by Bündner schist of the Grava- and Tomül nappes (swisstopo, 2005). The terrain features complex topography with a curved central fall path ending abruptly at the nearly horizontal but inter-experimentally changing riverbed, which also features fluvial terraces. The forest exhibits 593 mapped trees and consists mainly of European beech (*Fagus sylvatica* L.) and Norway Spruce (*Picea abies* (L.) H. Karst). Some European ash (*Fraxinus excelsior* L.), sycamore maple (*Acer pseudoplatanus* L.) and Scots elm (*Ulmus glabra* Huds.) trees and one European yew (*Taxxus Baccata*) tree, all with a diameter at breast height (DBH) $\leq 8$ cm were present in the acceleration zone ($\alpha_1$ in Fig. 1b). Mainly spruce trees were affected by the storm *Burglind* in the year 2018, leaving two distinct lying deadwood clusters behind including broken, and overturned trees comprising root plates (see the shaded areas in Fig. 1a). The upper cluster is located at 815 m a.s.l, slightly to the orographic right of the central dropping path, and it comprises nine deadwood piled logs with a basal diameter $d_0 \geq 20$ cm. The lower, main deadwood cluster spans almost the entire width of the experimental slope and comprises 20 logs ($d_0 \geq 20$ cm). Most of the logs were sap-bearing fresh wood (class 1, according to Lachat et al. 2013), but there were also a few sapless hardwood logs (class 2). Experimental logistics required the identification of relevant stems that could be re-installed for large mass experiments. Three high-resolution, UAV-based lidar missions were carried out to fully map the forest structure. During the first mission with a Riegl VUX-1UAV, we achieved a mean pulse density of 2256 m$^{-2}$ (Fig. 1c). The second mission was conducted with a Riegl MiniVUX-2UAV and a mean pulse density of 1973 m$^{-2}$ (Fig. 2a). The last mission was carried out with a DJI Zenmuse L1 sensor, resulting in 443 m$^{-2}$ pulses on average.

**Table 1.** Overview on the released rocks, subdivided by rock shape (cubic $EOTA_{111}$ and platy $EOTA_{221}$), rock mass (200 kg - 3200 kg), and state of the forest (DW) deadwood, CLR: cleared deadwood section. Due to the small number of experimental runs, $EOTA_{2600kg}$ and $EOTA_{3200kg}$ are hereafter listed together as a single mass class $EOTA_{\geq 2600kg}$. Bold fonts indicate the reconstructed experimental runs, included in Fig. 4.

| | $EOTA_{111}$ | | $EOTA_{221}$ | |
| --- | --- | --- | --- | --- |
| | DW | CLR | DW | CLR |
| $EOTA_{200kg}$ | 12 | 8 | 11 | 12 |
| $EOTA_{800kg}$ | **10** | **10** | **10** | **13** |
| $EOTA_{2600/3200kg}$ | **2/2** | **2/2** | **3/3** | **3/3** |

## 2.2 Experimental design and instrumentation

We released a total of 106 artificial, sensor-equipped rocks into two different states of the mountain forest to directly compare the protective effect of lying deadwood: 53 experimental runs into the original state, including naturally windblown deadwood (DW) and 53 experimental runs into the cleared forest (CLR, Tab. 1). The rocks were all released from the same starting area, a $4\,\text{m}^2$ area marked with a black cross in Fig. 1a . Due to the rather channelized topography within the first few meters of descent, the starting point was regarded as identical for all reported trajectories. Two symmetric rock geometries according to the European Organisation for Technical Assessment (EOTA) (ETAG 027, 2013) were cast from steel-reinforced concrete (density $\rho = 2650\,\text{kg m}^{-1} \pm 3\,\%$), along with three mass classes. While the cubic $EOTA_{111}$ features equal axis lengths (Fig. 1.d, f, h), the two longest, orthogonal axes are twice as long as the shortest axis for the platy-shaped rocks $EOTA_{221}$ (Fig. 1.e). The mass classes - which include minor mass differences between the rock shapes - were approx. 200 kg ($\sim 0.08\,\text{m}^3$), 800 kg ($\sim 0.30\,\text{m}^3$), and $\geq 2600$ kg, which comprised blocks with 2600 kg ($\sim 0.96\,\text{m}^3$) and 3200 kg ($\sim 1.23\,\text{m}^3$). The released rock masses were increased with consecutive days of experimentation starting with the smallest mass class and the deadwood (DW) configuration. It was expected that the $\geq 2600$ kg rock mass class would cause significant destruction during the deadwood experiments, not only to the lying deadwood logs, which would have been acceptable, as they were removed for the following experimental runs in the cleared (CLR) forest, but also to the standing forest. Major damage to the forest could have compromised the subsequent comparison experiments under CLR conditions. To prevent this, the natural deadwood was removed after the DW 800 kg runs. After the CLR experiments for 200 kg, 800 kg, and $\geq 2600$ kg rocks, the most relevant logs were reinstalled for the DW campaign involving the largest rock mass class. The GNSS position of each deposition point was recorded with STONEX S400 and S800 devices using the Swiss coordinate system CH1903+_LV95 with decimeter accuracy.

Evolving versions of in situ StoneNode sensors (Niklaus et al., 2017; Caviezel et al., 2018, 2021b) were deployed over the course of nine experimental days carried out over 2.5 years. While previous versions already enabled measurements of up to $4000°\text{s}^{-1}$ rotational velocity and 400 g per axis with a temporal resolution of 1 kHz, the latest version included the most stable hardware and compiler setting (Mayer et al., 2023). The aluminum sensor cases were made waterproof through silicon sealing and/or o-ring enhancement (Fig. 1h). Previous experimental campaigns stressed the importance of including continuous visual

tracking, realized via a high-resolution single camera setup (Caviezel et al., 2021a) covering the entire slope, or a panning setup (Noël et al., 2022). Both options are unfeasible in a forested environment due to the obstructed line of sight by the trees. Hence, 14 GoPro Black 7 cameras, recording 59.94 frames per second in a 4K setting (3840 x 2160 pixels), were mounted anew for each experimental day along the fall path. Large storage (SD cards with 256 GB) and external battery packs (RAVPower RP-PB043) ensured run-times of up to 8 hours, granting some temporal flexibility for dealing with any disruptions or challenges, such as late helicopter crew arrival or difficulties in rebuilding the deadwood section.

During the first four experiment days, the focus was on establishing a reliable trajectory reconstruction method. For this purpose, rock masses of up to 200 kg were used. Transportation logistics were handled via a four-wheel-drive crane truck, releasing the rocks from its overhead, pneumatic arm and picking them up at the river bed – if the rocks traveled that far. Rocks that did not reach the riverbed had to be remobilized and rolled down by hand. As manual remobilization was not possible for larger rocks, transportation was achieved via a helicopter with the appropriate loading capacity: an Airbus H125 for the 800 kg blocks and an Airbus AS332 for the heaviest rocks and for the installation of the deadwood logs.

Two Decagon-GS1 soil moisture sensors at depths of 10 and 30 cm were placed between the two deadwood clustes (Fig. 1.a), measuring the volumetric water content with a sampling interval of 1.5 h for continuous long-term soil moisture classification. Acceleration sensors (MSR165) were installed on deadwood logs according to their availability. The trade-off between storage capacity and the highest possible write rate (ultimately 200 kHz) was mitigated by adjusting the triggering thresholds after installation on the logs in the field, based on the resultant acceleration from the factual resting position of each log.

## 2.3 Trajectory reconstruction

The observation and reconstruction of rockfall trajectories in forests is a demanding task (Bourrier et al., 2012). In a previous open-land study (Caviezel et al., 2019) two different trajectory-reconstruction methods were described: the labor-intense *a posteriori* impact mapping (AIM) method and the (semi-)automatic dense cloud reconstruction (DCR) method. For the latter, synchronous, stereo video image pairs were photogrammetrically analyzed. The determination of the rock position was achieved through color classifications, but robustness difficulties emerged over the course of an experiment due to deteriorating contrast. A possible solution might have been the use of a differential point cloud analysis with the M3C2 algorithm (Lague et al., 2013) as an option. No semi-automatic videogrammetry approaches were implemented in this study for three reasons:

1. The tested, pairwise installation of GoPro cameras did not provide the desired results regarding three-dimensional reconstruction capabilities of fast-moving objects with a sufficient depth of field. This is mainly because the built-in wide-angle lenses are not capable of identifying small, fast-moving objects.

2. The subsequently used Blackmagic Pocket Cinema Cameras pairs (4K and 6K editions) resulted in better reconstruction capabilities. These cameras had to be operated by people, however, and not enough safe camera positions were available and those fulfilling the safety criteria did not have the ideal field of view.

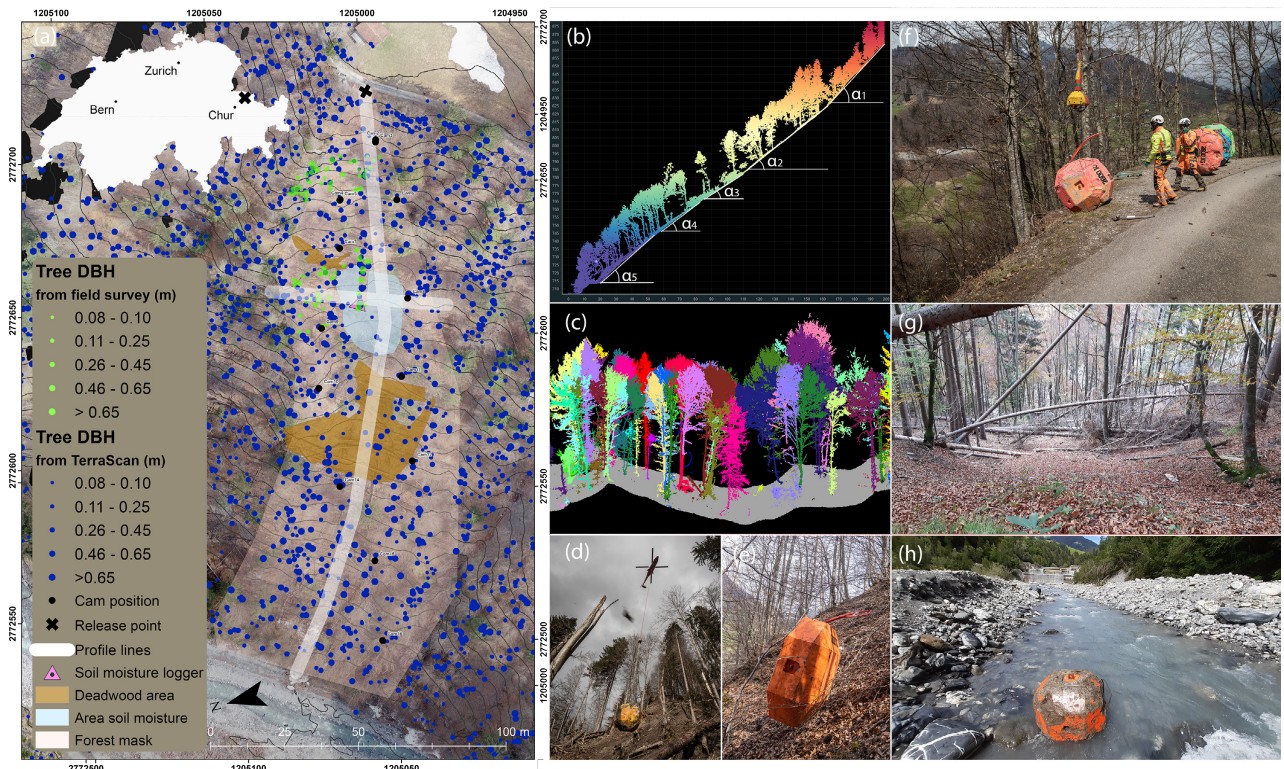

**Figure 1.** Overview of the test site *Schraubach*. (a) The release point area is marked with a black cross, the SE-NW oriented, longitudinal white line marks the central fall path on the slope, and the horizontal line marks a representative cross-sectional profile. The locations of single trees according to a detection algorithm (blue) and field survey (green) are depicted, along with the two deadwood clusters shaded in brown. Additionally, the camera positions and soil moisture comparison areas are indicated. (b) Forest visualization along the longitudinal profile, obtained from the pre-experimental lidar point cloud visualizing the stem-free area above the deadwood section and the slope angles $\alpha_1 = 48°$, $\alpha_2 = 39°$, $\alpha_3 = 33°$, $\alpha_4 = 38°$ and $\alpha_5 = 43°$. (c) Equivalent visualization but for the representative cross-sectional profile marked on panel (a). (d) Helicopter-assisted underload-slinging transportation of a cubic shaped-test rock. (e) acceleration phase of a platy-shaped rock. (f) Moment of the release of a 2600 kg rock at the release area, with the electronic hook being remotely triggered by the pilot. (g) Downslope view of the stem-free area above the lower, main deadwood section. (h) The longest run-out distances resulted in the rocks reaching the river *Schraubach*, highlighting the need for a waterproof in situ sensor. Photos (d), (e) and (f) by Matthias Paintner, SLF.

3. The desired depth of field of these cameras demanded a larger distance between each pair of cameras. As a consequence, more trees were present between the camera and the rock path, hiding the rock in either one or the other image. This tree shielding resulted in too few reconstructable points on for the moving rock.

Another promising method to retrieve rockfall trajectories with fewer manual inputs was described in Volkwein and Klette (2014), where a local positioning system (LPS) was established to track rock motions. The authors reported only the determination of two-dimensional coordinates in open-field experiments. While an extension to a fully three-dimensional position

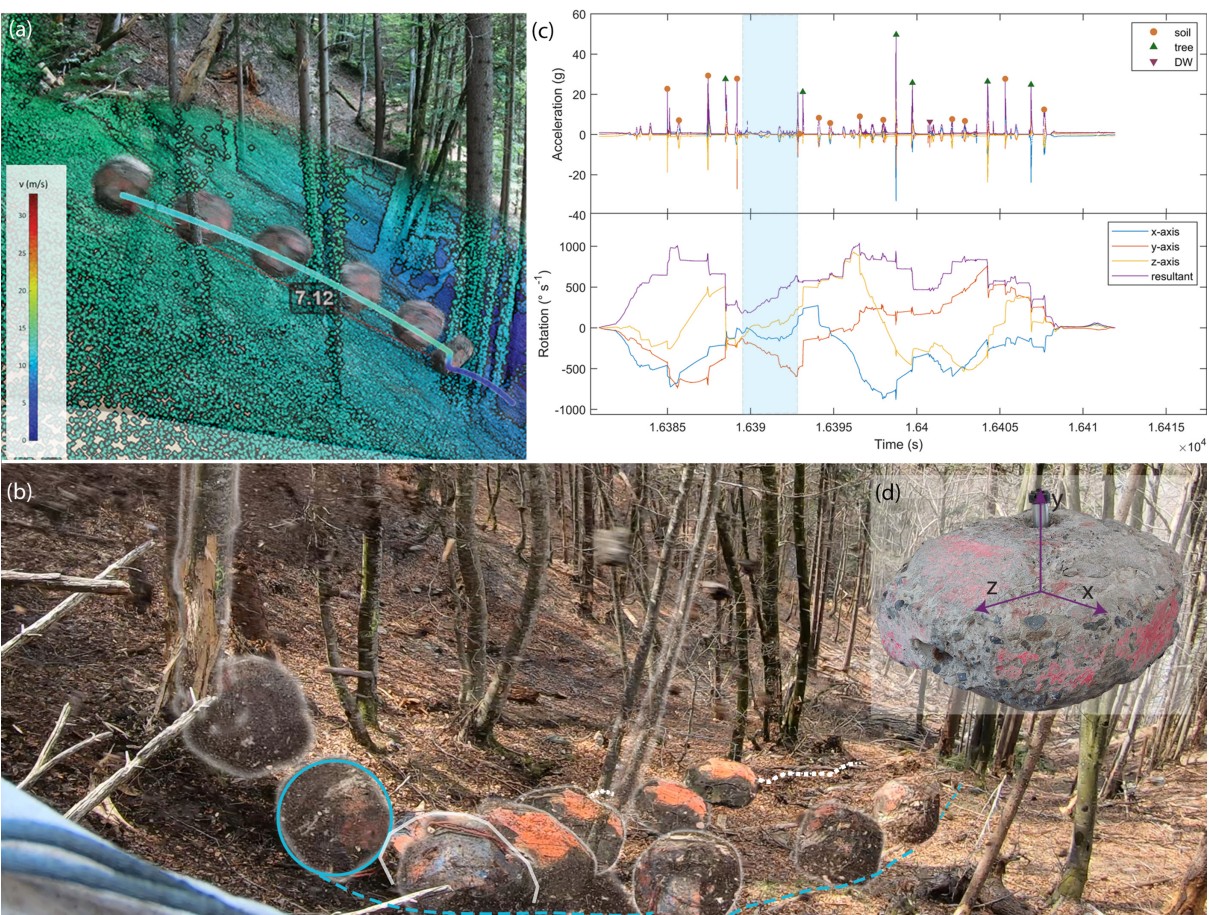

**Figure 2.** (a) Visualization of a reconstructed jump/tree–impact/jump succession in an image fusion of a video still and a lidar point cloud representation. The $\text{EOTA}_{221}^{800\,\text{kg}}$-rock jumps 7.12 m from lift-off to tree impact with a velocity change from roughly 12 m $s^{-1}$ to 2 m $s^{-1}$. Single jumps were reconstructed by combining video footage of the experimental runs, manually determined lift-off and impact positions thereof inside the lidar point cloud, and flight time measured in situ. (b) Work-flow visualization for rolling and sliding trajectory segments. The blue circle marks the assumed rock circumference, and the dashed line represents the manually determined rolling path. The white dotted line is the equivalent determination of a sliding path section. (c) StoneNode data for an exemple run visualizing the acceleration data used to identify jump times and rotational data, providing hints as to the state of kinematic regimes. Airborne phases are marked by free flight, i.e. zero acceleration besides earth gravity and flat rotational velocity curves while rolling phases normally feature a linear increase in the resultant rotational velocity combined with low but non-zero acceleration signatures marked as blue shaded area. The individual impacts are labeled according to their substrate classification. (d) StoneNode housing and the orientation of its axes within the borehole.

location might be feasible, a robust setup would require the installation of a transceiver inside the rock, where the signal strength diminishes drastically due to the EOTA rocks' steel reinforcement. Similar problems have been documented with

ultra-wideband (UWB) technology (Mayer et al., 2019) . These limitations prevented successful implementation within the scope of these experiments.

Due to these reasons, and hence missing a (semi-)automatic alternative, we applied an adapted AIM method for the reconstruction of the parabolic jump trajectories, fused with the recently proposed computer-assisted videogrammetric 3D trajectory reconstruction (CAVR) method (Noël et al., 2022). If StoneNode data were available, the timing of the jumps was based on these in situ measurements. The geographic locations of lift-off and impact were determined based on the footage and the georeferenced lidar point cloud displayed within the free, open-source WebGL based point cloud renderer potree (Schütz et al.,

2020; Schütz, 2021). Figure 2a shows the fused image of a video still, a potree lidar representation, and a reconstructed trajectory of a single tree impact. This is similar to the approach described by Noël et al. (2022). In contrast to the studies cited above, we reconstructed not only jumps but also rolling and sliding movements. The StoneNode data obtained in situ were additionally dissected for their content rather than just serving as a timer of the flight phases. Rolling phases usually show a linear increase within the resulting rotational velocity vector (blue area in Fig. 2c) – as opposed to rotationally invariant flying

phases. The StoneNode data (Fig. 2c) and the footage overlay (Fig. 2b) served as inputs for identifying and reconstructing the rolling trajectory segments.

     In addition to the reconstruction, every lift-off and impact position was classified according to the substrate present. The distinction between soft and hard forest soil impacts was defined based on the visibility of rock dust in the footage , arising from the weathered Bünder schists. The main impact categories and their sub-categories are listed below:

1. Forest soil (*soft*, *hard*, *loose rocks*)

      2. Tree impacts (*breakage*, *frontal impacts*, *lateral impacts*, *scratch*, *multi-stemmed* and *ramping*),

      3. Impacts on lying deadwood (*breakage*, *frontal impacts*, *scratch*, *root plate*),

      4. Impacts on standing deadwood, so called snags (*breakage*, *frontal impacts*, *scratch*),

      5. Scrub and small woody debris (*standing*, *lying*),

6. Snow

      7. Gravel (*riverbed*, *dam*).

     The workflow for deriving the rolling velocity is visualized in Fig. 2b. The rolling velocity was reconstructed using the maximum radius of the corresponding EOTA block, under the assumption of a slip-free rolling motion corresponding to the rotational data. The $x$-, $y$-, and $z$-coordinates of the trajectory were calculated along the hand-edited path observed from the

footage. The comparison of the rolled distance on the edited path and the lift-off velocity of the subsequent jump served as a quality check. To match these two constraints, an enhancing factor, usually between 10% and 20%, had to be included in the retrieved velocities. Sliding velocities of flat-angled but still moving $EOTA_{221}$ blocks were obtained by integrating the $x$ and $z$ acceleration data. The $y$-components were neglected, as they faced along the borehole (see the mounting of a sensor within an

EOTA$_{221}$ rock in Fig. 2d) being irrelevant for the direction of motion. The coordinates were draped along a hand-edited line, integrating the obtained velocity a second time to retrieve the real sliding distance. Figure 2c showcases an exemple run with the described impact classification for every impact. This information was assigned to every reconstructed trajectory alongside the complete set of temporally resolved parameters of interest for single rockfall trajectories, such as position coordinates, translational and rotational velocities, jump heights and lengths, and their according energies.

Due to the complex topography with a curved slope, the abrupt transition at the nearly horizontal base of the slope, the changing river bed within the experimental period, and the presence of fluvial terraces, the mean deposition altitudes (MDA) were analyzed instead of the mean run-out distance. Besides enabling the evaluation of deposition patterns, the reconstructed trajectories facilitated the comparison of all further parameters of interest between the two states of the forest and between the two rock shapes. We conducted such comparisons on the slope scale, but also within 10 evaluation screens. The latter are small-scale analysis areas with an altitude difference of 5 m. The trajectory reconstruction was accomplished for rock masses $\geq 800$ kg.

## 2.4 Impact analysis

Our analysis of the effects of rock impacts on forest elements involved both standing trees and fallen deadwood. To quantify these effects, we focused on the incoming rock energies associated with classified frontal impacts (FIs), and assumed a linear relationship between this energy and the cross-sectional area (CSA) of the wood. To ensure that our analysis was accurate and comprehensive, we set the y-intercept to zero. This approach allowed us to introduce different absorption coefficients $a$ for the observed species and wood condition within the energy absorption relationship:

$$E_{\text{abs}} = a \cdot \pi \left( \frac{DBH}{200} \right)^2 \tag{1}$$

## 2.5 Rockfall Simulations

Nowadays, three-dimensional rockfall simulations are the common tools used in practice to produce rockfall hazard maps and to dimension protective measures against rockfall. Such models need calibration and validation of the model parameters, either based on past rockfall events or rockfall experiments in order to deliver realistic results. Due to the larger data depth and the existing knowledge of the variability between single rockfall runs, the calibration of experimental data is the gold standard. In this study, we used RAMMS::ROCKFALL simulation program (Leine et al., 2014, 2021; Lu et al., 2019), which entails input capabilities for single trees in the forest representation (Lu et al., 2020) and has recently been enlarged and tested for dead-wood configurations as rigid three-dimensional truncated cones (Ringenbach et al., 2022c). The base scenario building for the RAMMS::ROCKFALL simulations contrasted a more time- and cost-intensive method with a more generic, automatized input generation. The resulting scenarios are subsequently labeled as close-to-reality forest (CRF) and generic (GEN) scenarios. For CRF, the commercial TerraScan software (Soininen, Arttu, 2021) was used to extract the DBH and height of single trees from

the pre-experimental lidar point cloud. Although some limitations for split trunks and multi-stemmed trees were registered, overall, the derived spatial distribution of the trees agreed well within the two observation plots (Fig. 1a). GEN comprised a randomly generated, generic forest composed according to the forest stand quantification, and required estimates of the number of trees per hectare, the mean DBH, and the DBH standard deviation for the specified tree area as input parameters. Generated tree positions were randomly distributed across the specified shape file. The deadwood representation was treated equivalently, using manually measured and digitized deadwood representations for CRF and deadwood configurations originating from the automatic deadwood generator (ADG, presented in Ringenbach et al. 2022a) for GEN.

For the CRF simulation scenario with deadwood, 31 logs with a mean base diameter $D_0 = 39.3 \pm 16.1$ cm were included, i.e. they were all above the consideration threshold of $D_0 \geq 20.0$ cm. This threshold was derived from observations, as smaller logs were often rolled over or broken. Their energy absorption capacity was calculated according to Equation 1 and $a_{DW}$ as stated in section 3.4. While the deadwood was removed removed from the simulation, most of the associated root plates were left in place. As some of these discs – made out of the soil and ground material formed by the root system – had an observed influence on the rockfall trajectories, we included the five most important root plates into the simulations (see white obstacles in Fig. 9b). They were extracted from the pre-experimental point cloud and assigned an ad-hoc absorption energy of 1500 kJ.

In this way, a forest mask comprising the area of the main trajectory paths was defined in which the necessary metrics for the forest generation within RAMMS were assessed. This process resulted in 580 trees ha$^{-1}$ with a mean DBH = 30.0 cm $\pm$ 23.3 cm. The $x$- and $y$-coordinates of the stem base, $D_0$, and the tree height $h$ served as input for the ADG. The tree height was estimated using the locally adapted ratio $h = \text{DBH}^{\frac{1}{1.1}}$. Additional input parameters were the mean wind direction with its standard deviation ($225 \pm 45°$) and the same root plate ratio as in the observed forest (16.1 %).

Based on the deposition pattern, the reconstructed translational velocities, and the distribution from the 13 reconstructed EOTA$_{221}^{800\ \text{kg}}$ trajectories, the mechanical soil strength parameter $M_E$ and the soil drag value $C_d$ were manually calibrated within RAMMS::ROCKFALL. It was aimed to reproduce a maximum run-out distance over the river, several deposition points in the adjacent terrain unit, and maximum velocities $\gtrsim 30$ ms$^{-1}$. The in-depth calibrated parameters from another rockfall experiment in forests (Ringenbach et al. (2022c), $M_E = 2.0$ and $C_d = 2.9$) were incrementally adjusted, resulting in overall forest soil parameter values of $M_E = 5.5$ and $C_d = 1.2$. These same soil parameters were used for the simulations of all rock shape and mass classes and all four forest configurations, including CRF$_{\text{DW/CLR}}$ and GEN$_{\text{DW/CLR}}$. A 2-meter line consisting of ten starting points acted as simulation release input, to represent the observed variations in the experimental starting point.

## 3 Results

### 3.1 Deposition pattern and the influence of deadwood

The complete deposition patterns for all the 106 test blocks, distributed across the two shape classes and all the weight classes, are shown in Fig. 3 for the forest state with lying deadwood (DW) and for the cleared forest state (CLR), along with their mean geographic center. Deadwood reduced the mean altitude (MDA) for five of the six shape-mass classes in the DW state

compared with the CLR experiments (Fig. 3). The MDA reduction order was as follows:

$$\Delta_{\mathrm{MDA}}^{\mathrm{EOTA}_{111}^{200\ \mathrm{kg}}} = -57.1\ \mathrm{m} \ > \ \Delta_{\mathrm{MDA}}^{\mathrm{EOTA}_{111/221}^{800\ \mathrm{kg}}} = -34.1\ \mathrm{m} \ > \ \Delta_{\mathrm{MDA}}^{\mathrm{EOTA}_{221}^{\geq 2600\ \mathrm{kg}}} = -15.3\ \mathrm{m} \ > \ \Delta_{\mathrm{MDA}}^{\mathrm{EOTA}_{111}^{\geq 2600\ \mathrm{kg}}} = -9.9\ \mathrm{m}$$

with the $\mathrm{EOTA}_{221}^{200\ \mathrm{kg}}$ shape–mass class being the sole exception, with an insignificant, minuscule difference between the two forest states of $\Delta_{\mathrm{MDA}} = -0.13\ \mathrm{m}$.

Another trend is visible in the deposition patterns and jump heights: platy-shaped rocks feature a shorter run-out distance in five out of six cases than their cubic-shaped counterparts of equal mass, irrespective of the forest state. This finding was corroborated by a greater MDA. Again, the $\mathrm{EOTA}_{221}^{200\ \mathrm{kg}}$ class was an exception from this trend, with a slightly longer run-
255 out ($\Delta_{\mathrm{MDA}}= 5.0\ \mathrm{m}$) than for $\mathrm{EOTA}_{111}^{200\ \mathrm{kg}}$. The complete distribution of the MDA is further compared to the simulations in Subsection 3.6

## 3.2 Four-dimensional trajectory reconstruction in a forest environment

In total, 63 rock trajectories were reconstructed in four-dimensional space, visualised in Figure 4. This exhaustive reconstruction complemented the static information content from the deposition patterns with kinematic information across the entire
slope. Table 2 highlights the median gyroscopic data, categorized according to the rock masses, shape, and the state of the forest. Figure 4a,b contrasts the DW versus the CLR state for the cubic $\mathrm{EOTA}_{111}$ blocks featuring blue markers, while the equivalent information for the platy $\mathrm{EOTA}_{221}$ is visualized in Figure 4c,d with magenta markers. Figure 4e zooms into the rolling and jumping motion of two example trajectories. The rolling trajectory section is the same as plotted in Fig.2a, that is the rolling part after a tree impact. The black crosses represent the identified contact points with assigned time stamps, as
described in the reconstruction workflow. The extents of the crosses represent the uncertainty ranges for each impact location corresponding to the spatial accuracy accumulated in the presented workflow. The succession of oblique jumps highlights the energy dissipation upon rock-ground interaction with its subsequent accelerated airborne phase.

Only after reconstruction did the parameters of interest (POI), and their change due to the removal of deadwood, become tangible. The trajectories are color-coded according to the translational velocities to make the data set visually comprehensible.
Velocities range up to $36.8\ \mathrm{m\ s}^{-1}$, where velocities $\geq 31.6\ \mathrm{m\ s}^{-1}$ are considered as outliers under the assumption of a normal distribution (Fig. 4h). By removing the deadwood, the mean velocity $\bar{v}$ increased significantly (T-test, significance level $\alpha = 0.01$) by 17%, from $12.5 \pm 6.0\ \mathrm{m\ s}^{-1}$ to $16.6 \pm 6.1\ \mathrm{m\ s}^{-1}$, while its standard deviation remained almost constant. Consequently,

**Table 2.** Median rotational velocities: gyroscope data streams from measurements made in situ for each mass class and EOTA rock shape, for the deadwood (DW) and cleared (CLR) forest states. The corresponding data from Caviezel et al. 2021a (Cav2021) serve as a comparison.

| | | 3200 kg | | 2600 kg | | | 800 kg | | | 200 kg | | |
|---|---|---|---|---|---|---|---|---|---|---|---|---|
| | | DW | CLR | DW | CLR | Cav2021 | DW | CLR | Cav2021 | DW | CLR | Cav2021 |
| $\mathrm{EOTA}_{111}$ | $(°\ \mathrm{s}^{-1})$ | 672 | 565 | 467 | 768 | 678 | 892 | 1008 | 586 | 1430 | 1485 | 1382 |
| $\mathrm{EOTA}_{221}$ | $(°\ \mathrm{s}^{-1})$ | 365 | 383 | 514 | 506 | 348 | 731 | 880 | 935 | 1122 | 1174 | 862 |

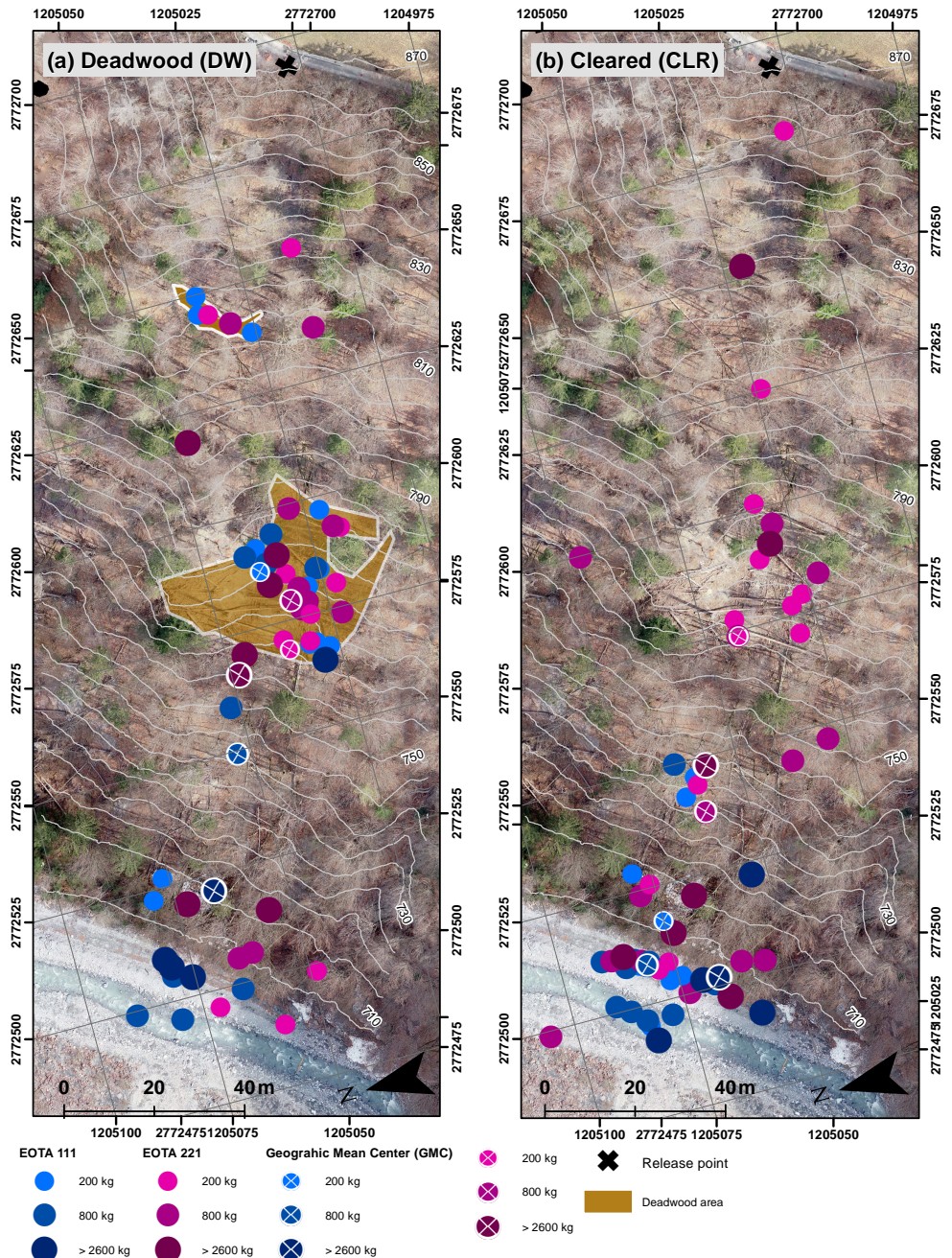

**Figure 3.** Deposition locations of all 106 rocks for the two different forest states (a) with lying deadwood logs and (b) after clearing of the deadwood logs. Blue markers represent the deposition points of cubic $EOTA_{111}$ rocks varying from 200 kg (light blue) to > 2600 kg (dark blue). Magenta markers represent the deposition points of platy $EOTA_{221}$ rocks varying from 200 kg (light magenta) to >2600 kg (dark magenta). The respective geographic mean centers for each rock category are indicated with a circle in the same color but marked with a cross. The deposition pattern of the $EOTA_{221}$ rocks stretches along the whole slope, compared to those of the $EOTA_{111}$ rocks.

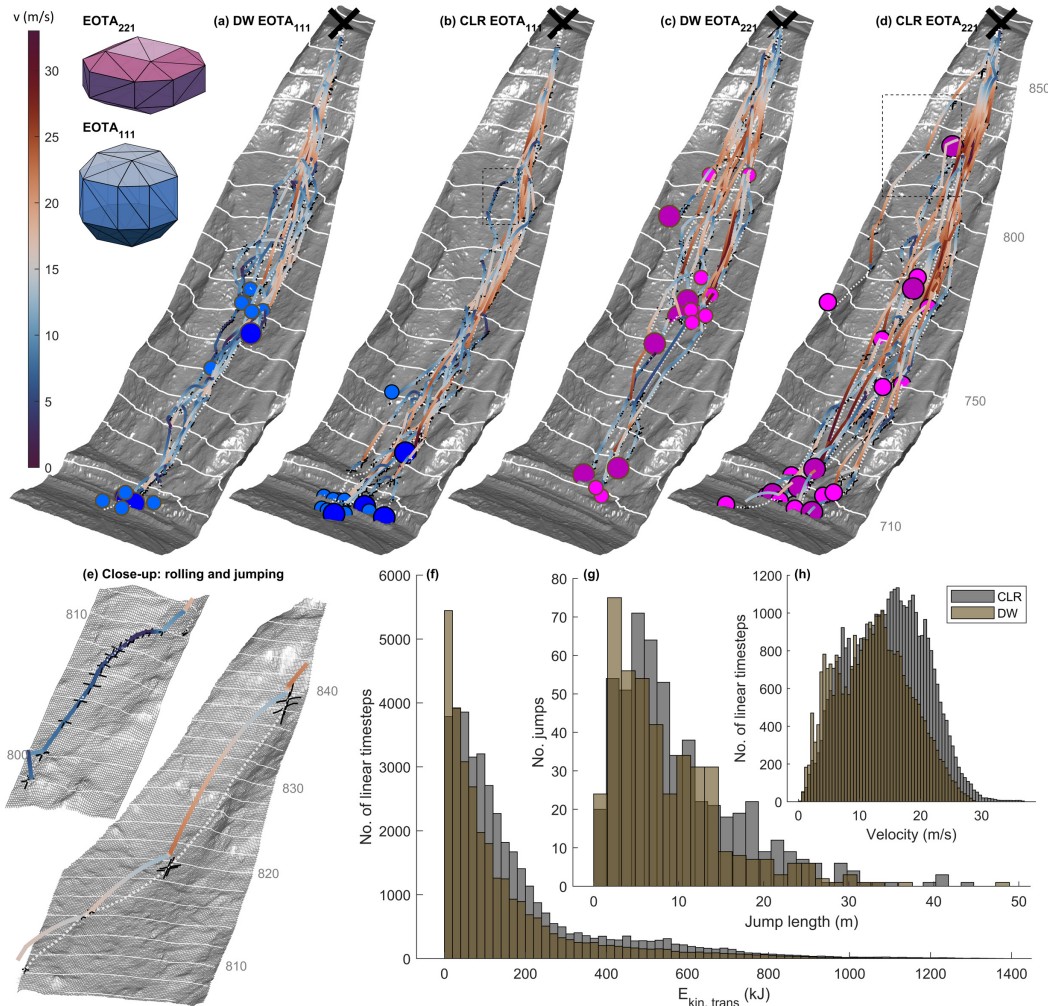

**Figure 4.** Velocities and deposition patterns of the 63 reconstructed rockfall trajectories: for the cubic $EOTA_{111}$ rocks in (a) the forest state with deadwood (DW, n=14) and (b) the cleared forest state (CLR, n=14), and for platy $EOTA_{221}$ rocks under (c) the DW (n=16) and (d) the CLR (n=19) site conditions. While the trajectories are color-coded according to the translational velocity, the same color coding as in Fig. 3 is used for the deposition points. The close-up views in (e) show the framed areas in (b) and (d). These enlarged displays feature a rolling trajectory (top) and a jumping trajectory (bottom). The rolling trajectory section visualizes the rock behavior after a tree impact. The succession of oblique jumps highlights the energy dissipation upon each rock–ground interaction with the subsequent accelerated airborne phase. The black crosses in (e) illustrate the identified contact points and their extent represents the uncertainty ranges for the impact locations. The uncertainties play a reduced role on the slope scale (a-d), as the crosses at every contact point are hardly noticeable any longer. The histograms highlight differences between the two forest states regarding (f) translational kinetic energy, (g) jump length, (h) and translational velocity for the two forest states.

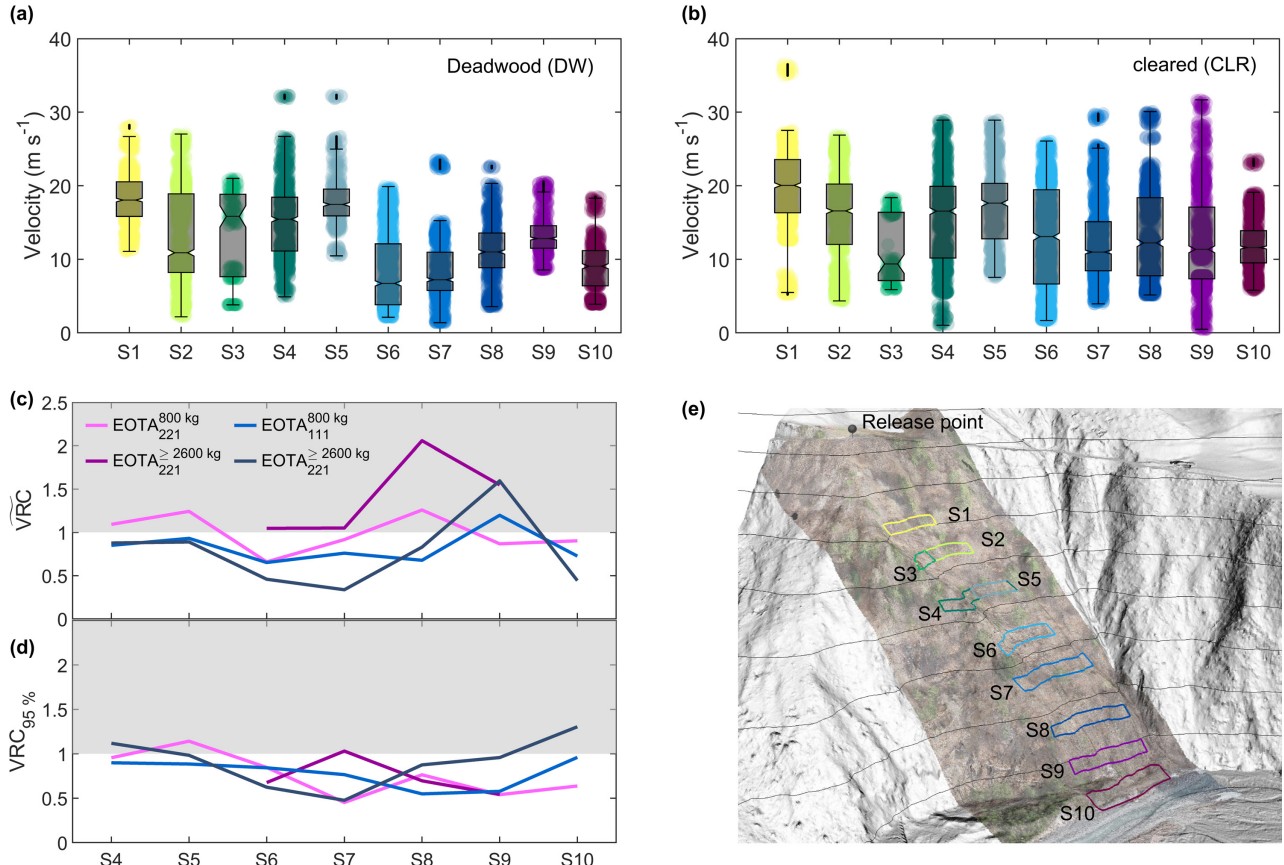

**Figure 5.** In-depth velocity comparison between the rock velocities derived from the experiments featuring forest states with deadwood (DW) present and with a cleared windthrow area (CLR). Boxplots of estimated boulder velocities for (a) DW and (b) CLR for the ten evaluation screens as shown in (e). All of these screens comprise a height difference of 5 m. Except for S3 and S5, which serve as control-screens to evaluate the effect of the upper deadwood cluster, all screens span the whole slope. (c) Median velocity reduction coefficient $\widetilde{\mathrm{VRC}}$, calculated as the median velocity per screen, rock shape and mass class for the DW forest state divided by its counterpart for the CLR forest state. (d) Comparison of the ratio of the $95^{\mathrm{th}}$ VRC percentiles, labeled as $\mathrm{VRC}_{95\%}$. In (c) and (d), 2600 kg and 3200 kg rocks are aggregated into one class $\geq 2600$ kg.

the translational kinetic energy values $\mathrm{E}_{\mathrm{kin,trans}}$ increased significantly (T-test, $\alpha = 0.01$) due to the change in horizontal forest structure (Fig. 4f). Also the mean translational kinetic energy $\overline{\mathrm{E}}_{\mathrm{kin,trans}}$ increased significantly by 19.3%, from 143 kJ to 171 kJ (T-test, $\alpha = 0.01$). The percentage increase was even greater for the median translational kinetic energy $\tilde{\mathrm{E}}_{\mathrm{kin,trans}}$ (+38.5%), although the absolute values were slightly lower (77 kJ to 106 kJ). The median of the reconstructed jump lengths (DW: $n = 424$, CLR: $n = 547$) increased by 11.8%, from 6.9 m to 7.7 m, with deadwood removal (Fig. 4g). The increase in the $75^{\mathrm{th}}$ percentile of the jump lengths was even more remarkable, rising by 18.8% from 11.7 m to 13.9 m.

Among the evaluation screens S1 to S10, the main difference in velocity between the DW and CLR states of the forest was observed in screens S6 through S10 (Fig. 5a,b). These screens comprise the main deadwood cluster (S6), the slope section below it (S7- S9), and reach down to the river (S10). Within all these screens, the 75th percentile velocities, shown as the upper box boundaries, were lower for DW than for CLR. The velocity reduction coefficient (VRC), defined as the ratio between the median values and the 95th percentiles observed in screens S4–S10 is compared in Fig. 5c and d, split according to rock mass and shape. While both sets of curves confirmed approximately equal entry velocities at the top of the lower deadwood section (S4 and S5), the median value $\widetilde{\mathrm{VRC}}$ was more volatile than $\mathrm{VRC}_{95\%}$. The latter primarily remained $<1$, with only $\mathrm{EOTA}_{111}^{\geq 2600\ kg}$ in S10 diverging, meaning that the highest velocities were achieved during the CLR experiments.

## 3.3 Impacts on standing trees

A total of 164 observed impacts on standing trees led to 44 tree breakages, while no uprooting was observed. The remaining 120 impacts were mainly frontal ($n = 47$). Lateral (28) and scratch (21) impacts were observed more often than impacts ramping over the stem base (13) and impacts on multi-stemmed beech trees (11). However, these numbers were reduced by 30–40% if only impacts with a minimum preceding airborne flight phase of 0.25 s were considered. This restriction ensured that only well-defined rock–tree interactions were analyzed when the change in kinetic energy $\Delta \mathrm{E}_{kin}$ was calculated.

Estimating the maximum absorbed energy of living beech and spruce trees was a two-way process. On the one hand, the incoming rock energies of classified frontal impacts (FI) on surviving trees were plotted against the wood cross-sectional area (CSA). The highest survived impact energies were fitted based on Equation 1 and a 95 % confidence interval (Fig. 6a, b). On the other hand, the observed beech (Fig. 6c) and spruce (Fig. 6d) stem breaks were compared with the incoming kinetic rock energy. The lowest observed impact energies leading to breakages were fitted based on Equation 1 and plotted with the 95 % confidence interval. These two functions per tree species, representing the boundaries of the colored areas in Fig. 6e (beech, FS) and 6g (spruce, PA) transformed into the DBH domain, differed only in the absorption coefficient $a$ within the energy absorption relationship (Eq. 1): The factors used for survival and for breakage for the two tree species were $a_{\mathrm{surv}}^{\mathrm{FS}} = 6498$, $a_{\mathrm{break}}^{\mathrm{FS}} = 3328$, $a_{\mathrm{surv}}^{\mathrm{PA}} = 585$ and $a_{\mathrm{break}}^{\mathrm{FS}} = 1310$ ($[a] = $ kJ m$^{-2}$), with DBH measured in cm, resulting in absorption energies $\mathrm{E}_{abs}$ in kJ. A direct comparison of the curves of the two tree species shows that beech had a higher absorption capacity. An image sequence based on video stills of the highest-energy impact a tree withstood without breaking during this experimental series is shown in Fig. 6f. The $\mathrm{EOTA}_{221}^{3200\ kg}$ block, traveling at a velocity of 29.2 m s$^{-1}$, exhibited a translational kinetic energy of $1372 \pm 72$ kJ. This translational energy was supplemented by a rotational velocity $\omega = 1127$ °s$^{-1}$, resulting in 154 kJ of rotational energy and a total kinetic energy of $\mathrm{E}_{kin,tot} = 1542$ kJ. The impact lasted 0.320 s and had a maximum measured acceleration of 47.5 g. The resulting impact force of 1492 kN was accompanied by a highly visible trunk deformation of $\sim 36$ cm or $\sim 73.5$ % of the DBH (Fig. 6f). Both, the outgoing translational velocity, $v_{\mathrm{out}} = 7.8$ m s$^{-1}$, and the rotational velocity, $\omega_{\mathrm{out}} = 292$ °s$^{-1}$, represented a reduction of 74% with respect to pre-impact values. The total kinetic energy of the moving block was even reduced by 93%, or 1436 kJ, due to the non-linear influence of these main drivers, which led to the stopping of the rock after a short sliding phase. Although the tree sustained visible structural damage, it also withstood the immediately following $\mathrm{EOTA}_{111}^{3200\ kg}$ rock also shown in Fig. 2b, which remobilized the earlier deposited $\mathrm{EOTA}_{221}^{3200\ kg}$ in a Boccia effect.

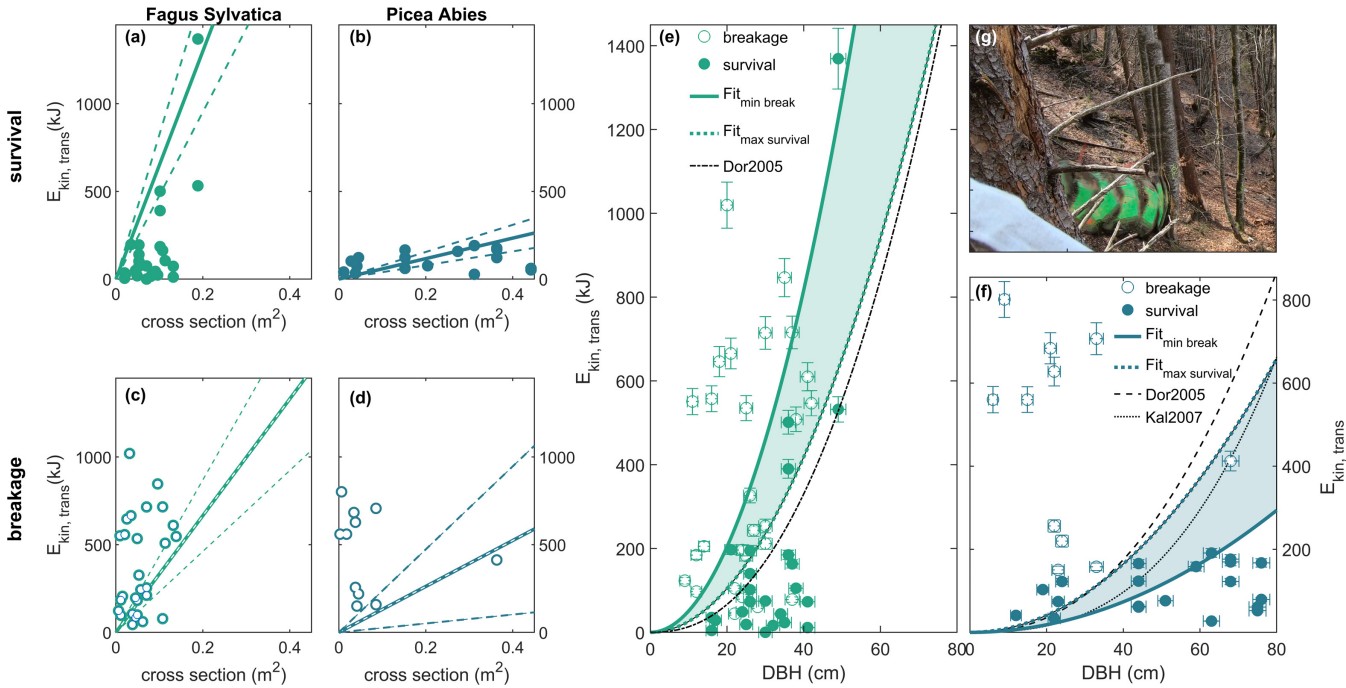

**Figure 6.** Relationships between $E_{kin,trans}$, DBH and the observed breakage of standing beeches (green) and Norway spruces (turquoise). (a) and (b) Tree survival: incoming rock energies of classified frontal impacts on surviving trees versus wood cross-sectional area. The fit is considered the upper boundary for survival. (c) and (d) Tree breakage: incoming rock energies of classified frontal impacts leading to tree breakage versus wood cross-sectional area. The fit is considered the lower boundary for breakage. (e) and (f) Tree diameter at breast height (DBH) versus incoming kinetic translational energy, with tree breakage and survival indicated by open and filled symbols, respectively. The lower and upper boundaries per tree are the linear fits from (a) and (b) transformed into the DBH domain. The existing correlations from Dorren and Berger (2005) and Kalberer et al. (2007) are shown for comparison as dashed black lines. (g) Video still sequence of the highest-energy impact a tree withstood without breaking during this experimental series.

This second impact on the tree with $E_{kin}^{tot} = 570$ kJ did not result in stem breakage, but rather in additional energy absorption of 433 kJ by the tree.

### 3.4 Impacts on deadwood

We observed a total of 55 deadwood impacts. Only direct, frontal impacts were analyzed further, as impact kinematics for more complex impact configurations become too difficult to resolve. For example, six breakages and two surviving logs were registered after an impacting rock first hit the soft soil immediately before the deadwood hit. This soil contact without a clear airborne flight phase before the deadwood impact made an accurate estimation of the incoming energy impossible. Hence,

a total of 27 traceable, frontal deadwood impacts were analyzed, with deadwood breakages resulting from the impact on 17 occasions (Fig. 7).

The relationship presented in Fig. 7 between the incoming rock energy and the maximum breakage energy was derived using functions similar to those used for standing tree breakage in Figure 6c and d. When adapted for deadwood, the absorption coefficient in Equation 1 was reduced to $a_{DW} = 415.7$. Most surviving deadwood logs had a diameter >49 cm. An example of a two-jump sequence of the impact onto a deadwood log without breakage is depicted in Fig. 7c. First, the $\text{EOTA}_{111}^{3200\text{kg}}$ broke a beech tree with DBH $= 38$ cm with its incoming kinetic energy $\text{E}_{\text{kin}}^{\text{tot}} = \text{E}_{\text{kin}}^{\text{trans}} + \text{E}_{\text{kin}}^{\text{rot}} = 508 + 59 = 567$ kJ, as pictured in the top right corner of Fig. 7c. The uprooting caused a loss of 75.7% of the rock's kinetic energy. The subsequent flight phase led to an energy gain to 369 kJ. After the rock–ground interaction shown in the center of Fig. 7c, the rock was left with only 71 kJ at the lift-off and arrived with a mere 82 kJ at the deadwood log. This low-energy rock was stopped by the 52 cm thick trunk. Thanks to the MSR acceleration sensor, directly mounted on the opposing side of the log, additional data on this impact were available (Fig. 7b). The data stream showed excellent agreement with the maximum rock accelerations $a_{\text{max}}^{\text{rock}} = 89.4$ $g$ and deadwood log $A$ acceleration $a_{\text{max}}^{\text{DW}_A} = 84.4$ $g$. The most impressive deadwood breakage happened during the first $\text{EOTA}_{111}^{2600\text{kg}}$ run after reinstallation of the major deadwood logs. This single run obliterated no less than the four logs visible in Fig. 7f. The presented multi-frame image visualizes, from left to right, the breaking of the logs $\text{DW}_G$, $\text{DW}_B$, $\text{DW}_F$ and $\text{DW}_C$. Reliable deadwood accelerometer data were available for two of these impacts Fig. 7e). The deadwood experienced significantly higher accelerations than the rock, i.e. $a_{\text{max}}^{\text{DW}_G} = 62$ $g$ versus $a_{\text{max}}^{\text{rock}} = 14.6$ $g$, and showed a highly divergent behavior, with $a_{\text{max}}^{\text{DW}_C} = 62$ $g$ versus $a_{\text{max}}^{\text{rock}} = 8$ $g$. An estimate of the energy reduction $\Delta\text{E}_{\text{kin}}$ for clean deadwood punctures, where the trajectory of the rocks was not significantly altered by the deadwood impact, is presented in Figure 7d. In this study, clean punctures were defined as having contact durations $\leq 0.5$ s with a single log, without preceding or subsequent soil contacts. Only six impacts complied with these criteria: five with spruce logs and one with a beech log.

## 3.5 Soil moisture

Experiments were exclusively held during dry weather conditions. Therefore, no abrupt changes in volumetric water content (VWC) of the soil were recorded within the time range of a given experiment. The soil moisture observation phase lasted almost 21 months, starting in mid-August 2019. The driest conditions were observed on 29 August 2020 with a VWC of 0.053 $\text{m}^3$ $\text{m}^{-3}$ at a depth of 10 cm, and with a VWC of 0.177 $\text{m}^3$ $\text{m}^{-3}$ at 30 cm depth. The highest VWC was measured on 21 August 2019 with a VWC of 0.340 $\text{m}^3$ $\text{m}^{-3}$ at a depth of 10 cm, and on 16 Mai 2020 with a VWC of 0.433 $\text{m}^3$ $\text{m}^{-3}$ at 30 cm depth. The measured $\text{VWC}_{30cm}$ and $\text{VWC}_{10cm}$ during the five experimental days with video coverage (vertical lines in Fig. 8a) were:

While the three experiments in 2019 and 2021 were carried out under rather moist soil conditions, the experimental days in 2020 featured rather dry soil conditions. For further analysis, one VWC value per experimental day was considered and paired with the available rockfall dynamics data from the impacts fully classified as soft-soil impacts located close to the soil sensor location, within the blue area shaded in Fig. 1a. The contacts needed to feature durations < 0.5 s. Robust accelerometer data were available for the 2020/2021 experimental days, and the identified accelerations ($n = 49$) were plotted in Fig. 8b. The mean accelerations followed a trend towards lower acceleration under wetter soil conditions, although this pattern was

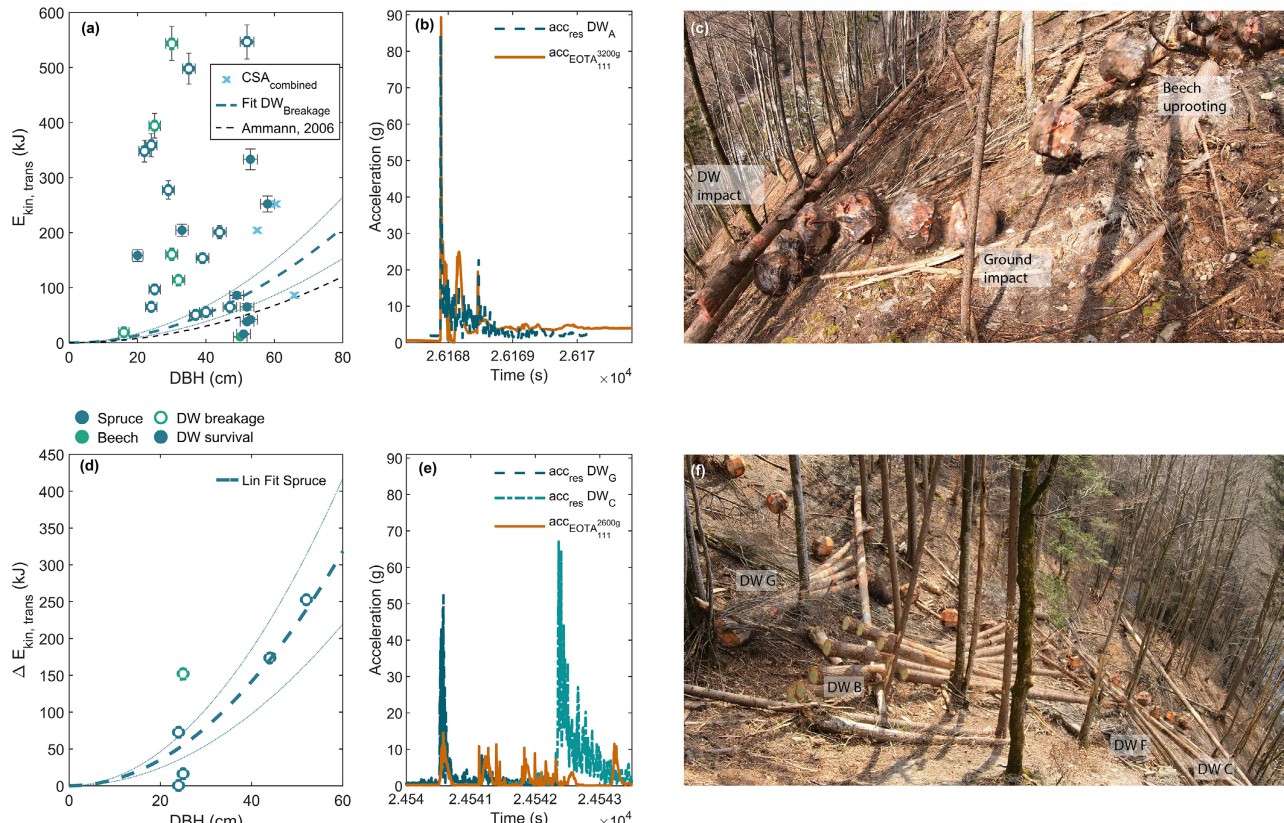

**Figure 7.** Relationships between $E_{kin,trans}$, DBH and the observed breakage of deadwood. (a) All 27 traceable deadwood impacts, with open symbols representing breakage and filled symbols indicating survival. The lower boundary of the breakage and its $95^{th}$ percentile are compared with the results of large-scale laboratory experiments (Ammann, 2006). (b) The congruent in situ acceleration stream of the $EOTA_{111}^{3200\ kg}$ and its opposing deadwood log $DW_A$. (c) Multi-frame image series of the $EOTA_{111}^{3200\ kg}$, visualizing a two-jump sequence leading to the final impact onto the deadwood log, corresponding to the sensor data shown in (b).(d) Estimate of the energy reduction $\Delta E_{kin}$ for clean deadwood punctures. (e) Sensor streams of an exemplary $EOTA_{111}^{2600\ kg}$ run causing several deadwood breakages showing significantly lower acceleration than the hit deadwood logs. (f) Corresponding trajectory section of the $EOTA_{111}^{2600\ kg}$ run for the acceleration window depicted in (e).

| Internal rockfall experiment number | Date | $VWC_{10cm}$ (m³ m⁻³) | $VWC_{30cm}$ (m³ m⁻³) |
|---|---|---|---|
| RF32 | 23.10.2019 | 0.227 | 0.346 |
| RF33 | 22.11.2019 | 0.256 | 0.363 |
| RF35 | 20.04.2020 | 0.188 | 0.302 |
| RF37 | 08.07.2020 | 0.138 | 0.256 |
| RF44 | 15.04.2021 | 0.230 | 0.373 |

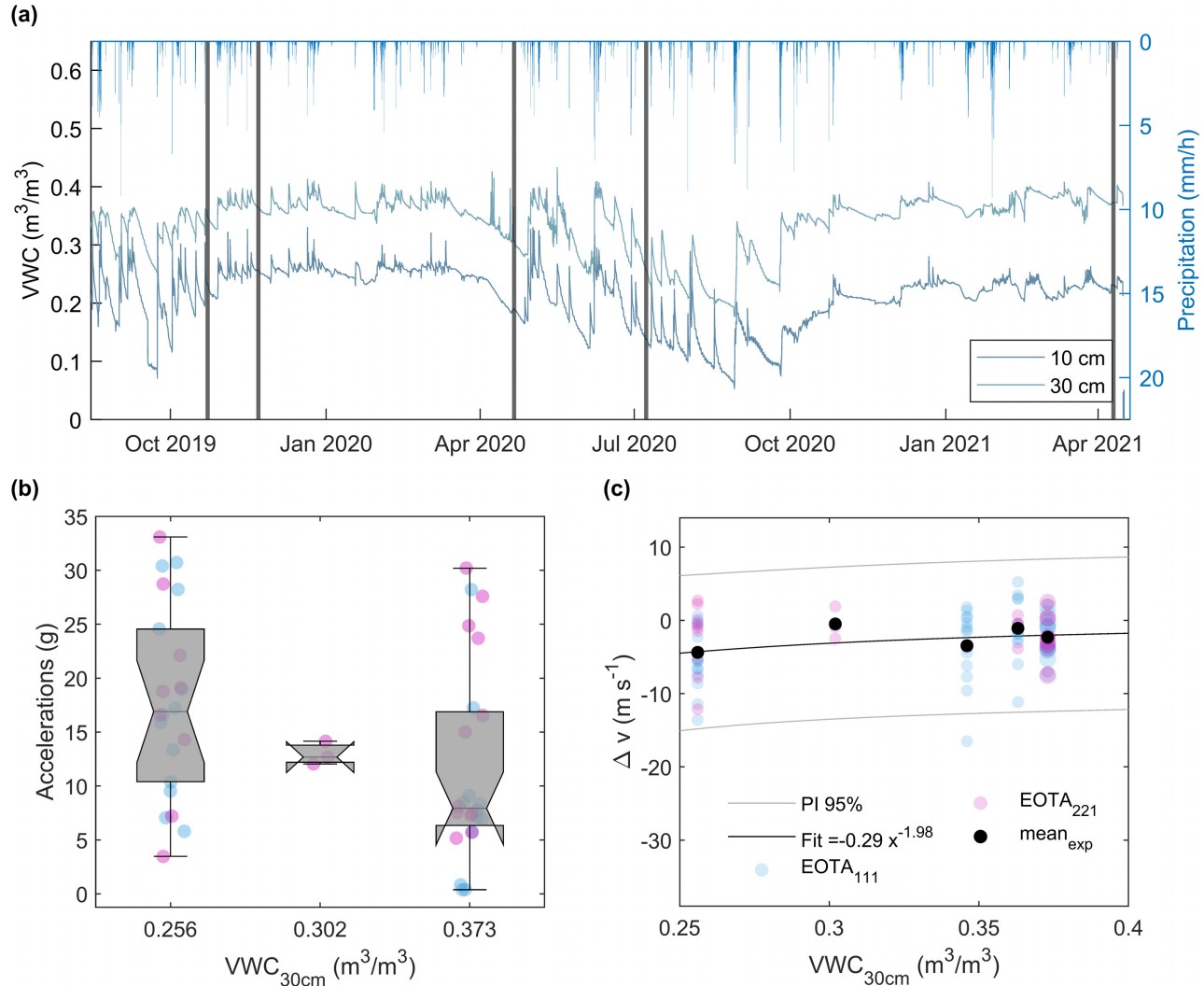

**Figure 8.** Soil moisture and its implications for rockfall impacts. (a) Volumetric water content (VWC) was measured at 10 and 30 cm depths over time compared with the precipitation measured at the closest MeteoSwiss weather station (4 km away, Federal Office of Meteorology and Climatology MeteoSwiss 2018). The vertical lines highlight the five experimental days with video coverage used for the trajectory reconstruction in Figure 4. (b) Accelerometer data versus VWC for soft soil impacts ($n = 49$). The median values of the measured rock accelerations of all representative impacts within the region of the soil moisture sensors show a weak negative correlation with the measured VWC. The partially overlapping notches of the three boxes indicate a lack of strong evidence for a significant difference between their medians at a 95% confidence level. (c) Velocity changes $\Delta v$ related to the observed VWC, with rock mass represented by symbol size ($n = 78$). The prediction interval (PI) is $\pm 10.6 \, \mathrm{m \, s^{-1}}$.

not statistically significant (ANOVA). Slightly more data points were available for velocity changes $\Delta v$ ($n = 78$; Fig. 8c). The
355 exponential relationship between $\mathrm{VWC}_{30cm}$ and $\Delta v$ is statistically not significant.

## 3.6 Simulations

Figure 9a and b visualize the RAMMS::ROCKFALL simulations performed for the $\text{EOTA}_{221}^{800\ kg}$ rock for the close-to-reality forest states (CRF) with lying deadwood (DW), $\text{SIM}_{\text{CRF}}^{\text{dw}}$, and after clearing (CLR), $\text{SIM}_{\text{CRF}}^{\text{clr}}$. The experimentally derived energy absorption factors for beech trees ($a_{break.}^{\text{FS}}$ = 3328) were used to determine the energy breakage limits for each tree. The equivalent images for the generic forest states (GEN), $\text{SIM}_{\text{GEN}}^{\text{dw}}$ and $\text{SIM}_{\text{GEN}}^{\text{clr}}$, are given in Fig. 9c and d. The $\text{SIM}_{\text{CRF}}^{\text{clr}}$ scenario served as a calibration scenario. Calibration was based on the agreement regarding run-out lengths and translational velocities, and in particular on successful replication of the trajectory paths veering towards the next natural terrain unit on the orographic right. The parameters of interest for the other scenarios were then compared with the reconstructed trajectories from Fig. 4. Veering-off trajectories were experimentally observed only under CLR conditions but occurred in the DW simulations as well. Scenario $\text{SIM}_{\text{CRF}}^{\text{dw}}$ showed a narrowing of trajectory paths below the main deadwood section from Fig. 9b to 9a, a feature absent in the GEN simulations.

Figure 10 compares the deposition altitudes from the experiments with the two simulation setups for DW and CLR forest states. The boxplots feature the 75th percentile as end of the box originating from its median and are again colored according to their weight and shape classes. The two grey shaded altitude bands indicate the locations of the two deadwood sections, with the lower, main deadwood section spanning from 785-760 m.a.s.l. The distributions of deposition altitudes differed strongly between the experiments and the simulations, but all medians were located within the main deadwood cluster, marked in gray in Fig. 10. The altitude dispersion in $\text{SIM}_{GEN}$ was more prominent than in $\text{SIM}_{CRF}$ and in the experiments. Deposition points for platy stones were distributed along the whole slope in the simulations without deadwood. However, the effect of the forest was overestimated, especially for large rocks, resulting in median deposition altitudes that were too large, especially for platy rocks.

Figure 11 depicts the mean rotational values $\bar{\omega}$ for the experiments and for the two simulation setups, again for DW and CLR scenarios. The results show strong congruence for all weight and shape classes under investigation, independent of the level of detail of the input forest. A decrease in rotational speed with increasing moment of inertia is clearly visible.

## 4 Discussion

Rockfall hazard assessments rely on the proper treatment of the interaction of the rock with any given opposing object. While the airborne phase is a mere oblique throw, with gravity as the lone acting force, the highly non-trivial impact mechanics render rockfall problems into an erratic, almost – at least for the layman – unpredictable process. Rock–ground interactions have drawn substantial attention in the civil engineering community, while research into the interaction with wood, particularly trees, lagged behind for a long time. Recently, interest in this interaction was renewed from an experimental (Olmedo et al., 2015) and numerical point of view (Toe et al., 2017). However, the experimentally examined scales were of more theoretical interest. Advances in dendrogeomorphological studies enabled the spatiotemporal analysis of past rockfall events based on tree damage (Trappmann et al., 2013). A wide range of studies in this area has contributed to an improved understanding of rockfall–forest interactions, better validation of rockfall trajectory models, and thus more accurate hazard assessments (Stoffel

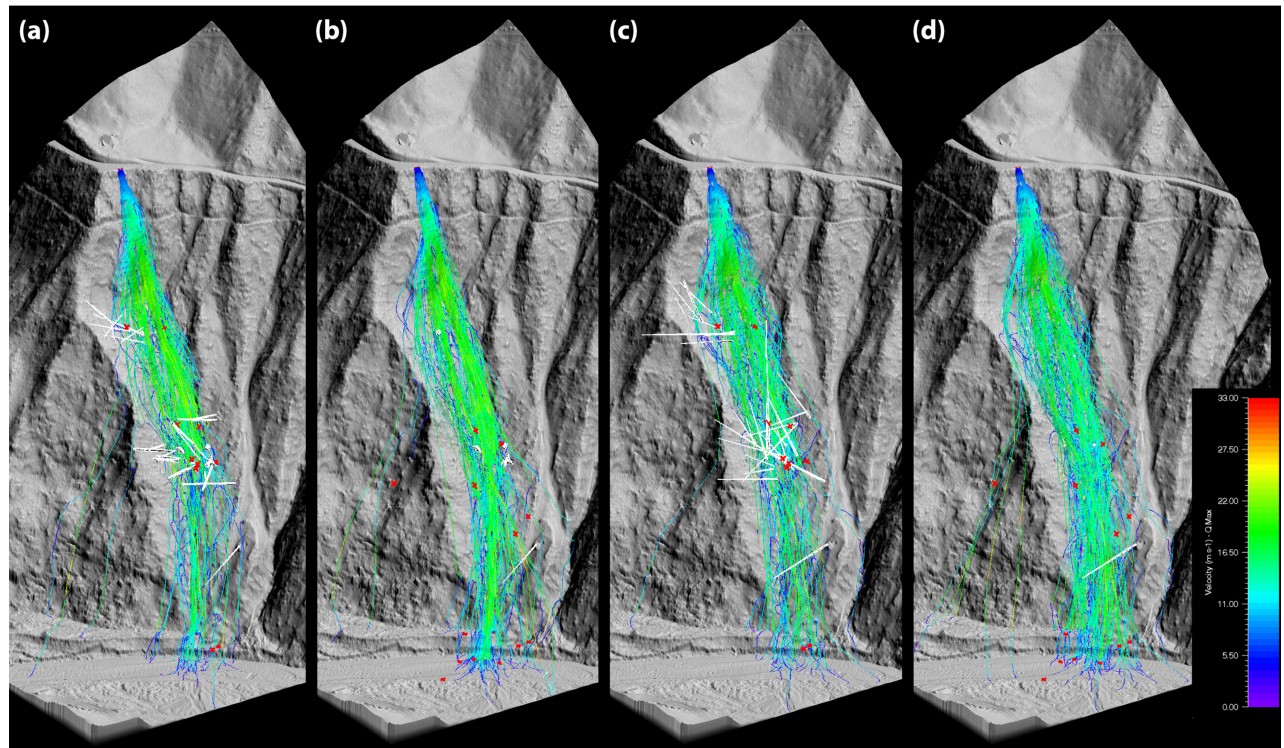

**Figure 9.** Visualization of the RAMMS::ROCKFALL simulations for $EOTA_{221}^{800\ kg}$ and their corresponding experimental deposition points (red crosses). (a) and (b) Close-to-reality (CRF) setup, with forest information extracted via commercial TerraScan software from a high-resolution lidar point cloud complemented with on-site, manually measured and digitized deadwood (white lines), for (a) the deadwood (DW) scenario and (b) the cleared forest (CLR) scenario, which served as a calibration scenario. (c) and (d) Generic scenario with standing trees randomly generated within RAMMS::ROCKFALL, based on estimates of the number of trees per hectare, the mean DBH, and its standard deviation, for (c) the DW scenario and (d) the CLR scenario. The deadwood in (c) originates from the automatic deadwood generator (Ringenbach et al., 2022a).

and Perret, 2006; Trappmann et al., 2014; Corona et al., 2017). However, these previous studies did not provide insight into the core of the temporally resolved rock–impact problem. Here, we presented a multi-year experimental campaign focusing on full-scale, single-block rockfall experiments in a forest with deadwood clusters. This setup made it possible to examine the intricate interplay of rock–ground–forest interactions and their implications for kinematics, by quantifying energy dissipation during these impacts. Full-scale experiments are inherently costly and complex, and rockfall experiments are no exception. The main advantage with respect to dendromorphological studies is the wealth of data originating from such an experimental campaign and the comprehensive, detailed reconstruction of rockfall trajectories, justifying the large resource allocation. For 63 trajectories, we reported a exhaustive data inventory comprising the complete set of kinematic parameters combined with detailed information about every impact with soil and tree hit configurations.

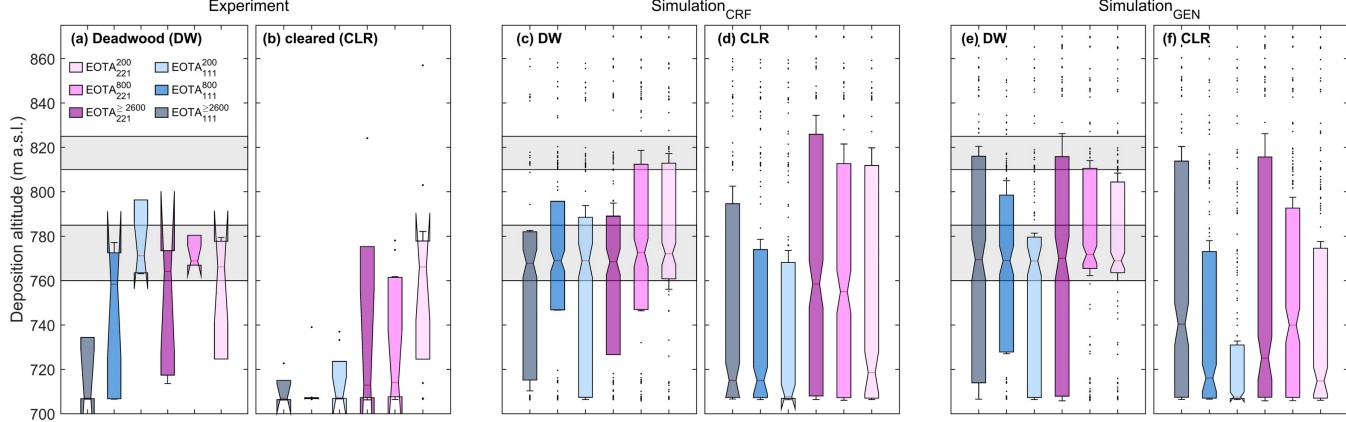

**Figure 10.** Summary boxplots of the block deposition altitudes for deadwood (DW) and cleared (CLR) forest states, expressed as elevation in m a.s.l. (a) Experiment DW, (b) experiment CLR, (c) close-to-reality (CRF) simulation DW (d) CRF simulation CLR, (e) simulation with generic forest setup and DW, and (f) GEN simulation CLR. The shaded areas depict the elevations of the two deadwood clusters, with the lower, main deadwood cluster spanning 785–760 m a.s.l. The boxplots feature the median values, with standard box sizes of one interquartile range (IQR), comprising 50% of all values. The boxes are colored according to their weight and shape classifications. The whisker length is compared with standard boxplots reduced to 0.1·IQR.

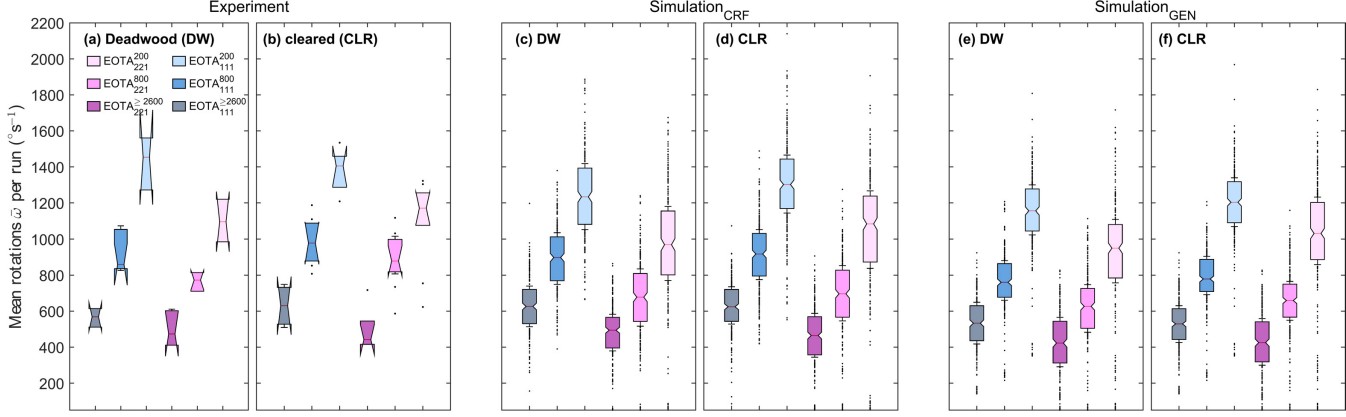

**Figure 11.** Summary boxplots of the rotational block velocities for deadwood (DW) and cleared (CLR) forest states. (a) Experiment DW, (b) experiment CLR, (c) close-to-reality (CRF) simulation with DW, (d) CRF CLR, (e) generic forest setup (GEN) DW, and (f) GEN CLR. The boxplots feature the median values, with standard box sizes of one interquartile range (IQR). The boxes are colored according to their weight and shape class classifications, and whisker length = 0.1·IQR.

## 4.1 A new facet of rock shape relevance

The experimental campaign, planned as a continuation of experiments in unobstructed terrain and hence featuring a strong focus on deadwood effects, yielded surprising new insight about a facet of rock shape relevance for rockfall propagation. In unobstructed terrain platy-shaped rocks tend to exhibit a significantly higher lateral spread than more compact cubic rocks, leading to an increased hazard potential. This shape-dependant hazard potential is reversed in forested areas. When platy-shaped rocks descend forested slopes, they feature generally shorter run-out distances than their mass-equivalent cubic counterparts. Over open land, their upright, wheel-like descending motion rotating around the shortest axis, featuring the highest moment of inertia, leads to wide, straight trajectories, often deviating from the line of steepest descent. In a forested environment, chances are high that this configuration is disturbed at every single tree impact, often resulting in a sliding phase on the rock's flat side. Due to higher friction, platy-shaped rocks are then prone to stopping completely. In contrast, the behavior of cubic rocks differs, as the rock geometry features a symmetrical axis without a preferred configuration of travel. After a tree impact, the rock starts to spin around another axis, easily remobilized downslope if the slope gradient is sufficiently high around the trees. While there is a lower probability of a tree hit for a platy-shaped rock than for a cubic rock, due to the smaller cross-section, the stopping effect of a single tree impact is higher for platy-shaped rocks. This shape effect is corroborated by comparing the rate at which rocks of the two shape classes passed the deadwood section. Under cleared conditions without deadwood, all rocks of the three cubic-shaped mass classes surpassed the former deadwood section, while a total of nine platy-shaped rocks were stopped in this section. This difference can be mainly attributed to the interplay of the larger frictional cross-section while sliding after a tree hit with the slightly lower slope inclination ($\alpha_3 = 33°$) and the increased soil roughness within the former deadwood section left by the clearing work. Although the inclination and roughness also affected EOTA$_{111}$-rocks, the sliding, platy-shaped EOTA$_{221}$-rocks were particularly impacted due to their lower center of mass.

## 4.2 Insights into rock–tree interactions

The exhaustive analysis of rock–tree interactions, for both deadwood and living trees, makes it possible to determine the minimum breaking energies for trees and deadwood of a given description, contributing valuable new data on this topic. For deadwood, for example, the comparison with values from large-scale laboratory experiments (Ammann, 2006) showed 1.7 times higher values for energy absorption capabilities in our real-world experiments. There are several reasons for the better performance of deadwood under realistic conditions. The entire natural impact configuration is dynamic, depending on the deadwood cluster, the exact geometrical impact direction and angle, and the freedom of logs to deflect and move skyward, where they had room and are not as likely to be trapped and clamped. Thus, the energy is dissipated over a greater distance and against gravity, both of which seem to increase the energy capacity of the trunk. Venturing away from the reproducibility of lab setups, the main drawback of real-world experiments is the large variability in boundary conditions. The manifold observed breakage mechanisms play a role in the observed variability in breakage energy: a log absorbs more kinetic energy while bending until the shear forces in the log cause it to break. However, this bending process, as with the initial impact, is strongly influenced by geometric conditions, such as jamming behind trees or the specific impact location of the rock with respect to the

deadwood log's clamping points. Furthermore, organic decay makes a log more brittle, which reduces the shear force capacity over time. As we observed rather fresh stems on the test site, the latter can be neglected in this case, although incorporating deadwood decay is an important issue in long-term forest management (Ringenbach et al., 2022a).

The data from our experiments can be used to further evaluate the energy absorption of living spruce and beech trees. Unlike in previous studies, we refrained from plotting a single relationship, hence a lower limit, between survival and breakage. We rather presented an area between the lower limit of breakage and the upper limit of survival, where increased breakage probability starts. We attribute the reported bandwidth of energy absorption capacities to different tree properties, stem rot, root size, tree vitality or soil depth (e.g. Toe et al. 2017). This range is also slightly higher than previously reported values from large-scale experiments (Dorren and Berger, 2005). The referenced value from Dorren and Berger (2005) is based on silver fir trees in a similar experimental setup, where impacts led to the breakage of nine trees. To estimate absorption energies for other tree species, conversion factors were proposed at that time based on small-scale experiments. This rescaling can now be verified with the results of the present study: The comparison line in Figure 6e shows that the present study expects even slightly higher absorption energies for beech.

For the derived breakage criterion for spruce, there are two comparison possibilities: As before, the values of Dorren and Berger (2005) – down-scaled from silver fir to Norway spruce – and the results of the field campaigns of Lundström et al. (2009) and Kalberer et al. (2007), which involved impacting trees with a wedge-shaped trolley along wire ropes, mimicking rockfall. Here the literature values tend to be at the upper edge of or even above our proposed area of increased breakage probability (Fig. 6f). The shift towards lower energies for spruce is probably due to the absence of high-energy impacts on thick spruce trees in our study. There was only one impact $> 400$ kJ on spruce with DBH $> 40$ cm. This particular tree had already survived three hits of 122 kJ, 170 kJ, and 176 kJ without complete breakage, but most probably the trunk suffered structural damage. Therefore, we attribute the lower energy absorption capacity to this sole supporting point of the fitted curve at higher energies, leading to lower confidence for high-energy behavior. Thus, higher literature values are justifiable. In addition to the visible damage caused by impact, the root systems of trees that remain standing may also be affected by mechanical damage. Our observations suggest that some impacts caused a substantial displacement of the trunk, resulting in deformation and shifting of the entire root ball. This displacement may have an effect on the structural integrity of the tree and its ability to withstand future disturbances, even if the tree did not break during the initial loading. Although the majority of the estimated displacements were less than 20 cm, one particularly large displacement is displayed in Fig. 6f, which was roughly equivalent to the diameter of the tree trunk (49 cm). As a result, future studies should consider whether such trees may be more vulnerable to rockfall or windthrow damage as a result of the impact-induced root damage.

## 4.3   From field observations to numerical use

The integration of living and dead trees into numerical models is critical to fully represent realistic and relevant boundary conditions of rockfall simulations in mountainous regions. The simulations performed here demonstrate the importance of incorporating capabilities to represent realistic forest configurations. They also confirm the trade-off between universally applicable generic solutions versus more sophisticated, field-based input assessments. While the extensive input data of the

close-to-reality (CRF) simulations performed better when directly compared with the experiment, generic solutions (GEN) can still serve as a powerful and rather inexpensive tool to expand those simulation capabilities to the regional scale. Nonetheless, the details of deadwood integration into numerical models have to be followed closely. A single, DBH-related energy threshold per standing tree might be enough, as most impacts happen in the lower, uniform section of a trunk, and its diameter at the impact location is usually reasonably close to its DBH. The situation is different for deadwood: the energy absorption capabilities differ greatly, depending on whether an impact happens at the deadwood base or close to the former tree top, where the diameter is small (in the low tens of centimeters). Future numerical implementations might need to accommodate this fact, either by detecting the diameter at the point of impact and adapting the energy threshold or by splitting the deadwood logs into several sections, each of equal diameter.

The forest rockfall retention capabilities were overestimated in our simulations, with the median deposition for large rocks located too far upslope. The accurate detection of tree diameter at impact locations, for both standing trees and deadwood, might rectify this shortcoming. Another explanation for the overestimation is the above-mentioned deficiency in the single-tree detection of multi-stemmed beech with the applied detection algorithm. Figure 1c suggests that most trees were detected correctly. However, when small branches fill the gaps between the individual stems of multi-stemmed beech trees between 1 m and 1.5 m above ground, it becomes almost impossible for the detection algorithms to depict reality. The consequences are greater for larger rocks than for smaller ones. Assuming a single beech is detected with a DBH$= 70$ cm, instead of three trees with DBH$= 30$ cm each, the kinetic energy required for its potential breakage is $\geq 1250$ kJ, equaling an impacting velocity of $v > 28$ m s$^{-1}$ for a 3200 kg rock. The necessary breakage energy for a single beech with DBH$= 30$ cm is roughly 235 kJ, meaning an impact velocity of 13 m s$^{-1}$ for a 3200 kg rock, which was often exceeded. For smaller rock masses ($m = 200$ kg), the algorithm detection problems are not significant. Realistically, already the 235 kJ required to break a single multi-stemmed beech is unlikely to occur, as an impact speed $\geq 48$ m s$^{-1}$ would be necessary.

Small rock masses $m = 200$ kg, on the other hand, were not sufficiently retained enough by the simulated deadwood. The simulated rocks reached the base of the slope too often. A possible explanation could be that the minimal effective deadwood diameter of 20 cm was too large for the smaller rock masses. Another reason could be that the branches of the deadwood, which were not included in the simulations, affected the rocks during the experiments. Despite this discrepancy, the simulations still give an outlook for the protective capacity of the deadwood in a few years, after the branches have decayed. Finally, to fully represent a deadwood cluster, the possibility needs to be considered that loosely placed deadwood logs with a smaller moment of inertia than the rock move freely upon impact or are pushed away without breakage, as observed in the experiments. In contrast, in the model the simulated deadwood logs acted as rigid bodies sticking to the ground and each other, leading to simulated breakage if the logs were thin. However, in some cases, the simulated trajectory ended due to an overestimated simulated friction between the stem and the ground.

### 4.4 Rockfall kinematics in forests

In agreement with the findings from previous experiments, the rock kinematics did not differ across shape and weight classes in unobstructed terrain. The steep acceleration section with almost no thick trees enabled a uniform acceleration phase for all

the test rocks. This manifests in velocity values that hardly diverged in the observation screens S1–S5. This is an important prerequisite for valid comparison concerning the deadwood effect. The median velocity changes between the deadwood (DW) and cleared (CLR) scenarios showed variances larger than the 95$^{th}$ percentile. Owing to the quasi-binary nature of rock–tree interactions, i.e. hit or no-hit, the effect on a single trajectory is highly divergent. A standing forest, and equally deadwood clusters, influence the rockfall trajectory in a stochastic manner. Current hazard assessments often rely on the 90–95th percentile as a cut-off criterion of rockfall simulations. The velocity decrease observed in the deadwood scenarios was more pronounced in this study (Fig. 5d). The sound incorporation of the impact physics and their effects on rockfall trajectories within the numerical models is an anteceding key requirement. This might get more important if the ongoing trend of hazard risk consideration continues. Such risk considerations incorporate the in general higher release probability of smaller rocks, which are in turn better retained by deadwood and other nature-based protection solutions.

The velocity reduction in lower-lying observation screens is clearly visible in the $VRC_{95\%}$ for the cubic-shaped rocks. This is yet another manifestation of the easier remobilization of cubic-shaped rocks in forests, as the $EOTA_{111}^{\geq 2600 \text{ kg}}$ rocks steadily increased their speed from screen S7, and the $EOTA_{111}^{800 \text{ kg}}$ rocks from screen S8. Such retrieval of kinetic energy was not observed for platy-shaped rocks, as the standing forest has a greater capacity to slow down and retain rocks of this shape. This demonstrates that the protective effect of deadwood rapidly decreased for cubic-shaped rocks with increasing distance below the deadwood cluster. The distance from S6 to S8 was 50 m, and the distance from S6 to S9 was 75 m. Most of the observed trajectories reaching the river level during the DW experiments followed a small, barely tree-stocked couloir. Therefore, a comparison with the maximum gap length for silvicultural measures (Frehner et al., 2005) is given. The proposed gap length of 40 m is slightly shorter than the observed distances, whereby this reserve is meaningful since the tree hit probability increased with the additional 10–25 m.

The presented in situ StoneNode data were an important aspect of the reconstruction methodology, and the values – although possibly site-specific – offer an easily accessible, high-precision quality check for any given rockfall simulation. The rotational data, in particular, represent highly accurate data streams, and congruence between experiments and simulations increases model plausibility. Engineering practice often relies on the qualitative reproduction of key features of the scenario under investigation but lacks the tools or the input data to proceed with more elaborate methods. Future calibration methods might include a fusion of machine learning algorithms with feedback loops on input parameters, such that the query for optimal model parameters could be additionally refined, a pathway in which the presented data set could be foundational. The site-dependence of rotational data can be assessed by comparing it with the corresponding data of Caviezel et al. (2021, Table 2). The two data sets feature a similar reciprocal dependence with the mass, or more specifically, with the moment of inertia. Considering the damping effect due to the tree hits, the generally higher values reported here compared with those from the open slope might be surprising. The greater overall slope inclination (measured along a 145 m planar distance) at the *Schraubach* site (39.6°) compared with that at the *Chant-Sura* site (36.3°) might explain this finding. For future $\bar{\omega}$ estimates, the inclination should be considered in addition to the moment of inertia.

## 4.5 Are rockfall dynamics related to soil moisture?

Despite the presence of a forest stand on the test site, roughly two-thirds of the classified impacts were rock–soil interactions. This ratio might be site- and forest-density-specific to a certain extent, but it equally reveals the importance of understanding rock–soil interactions within forests. The decreasing maximum acceleration per impact with increasing soil moisture can be considered intuitive: the wetter the soil, the softer its surface, and the lower the resistance experienced by the rock during soil penetration. This supports the observation of deeper scaring depths in wetter soils by Vick et al. (2019). The lower $\Delta v$ in wetter soils observed in our experiments is more surprising but is not a robust result. A reason for the non-robust results might be, that only one soil moisture measurement was taken or the already performed spatial crop of impacts to the area with presumed similar conditions was too wide. Altered soil parameters, due to changing climatic boundary conditions, might affect loading cycles according to Gerber (2019) and finally the run-out distances (Vick et al., 2019). We argue that the change in the intra-soil parameter range is negligible compared with the importance of overall correct soil attribution and accurate representation of rock kinematics within rockfall models. However, a single correct ground parameter setting is – owing to the natural variability of different soils and ground types – difficult to achieve. This is reflected by the slightly adapted parameters used to depict the observed "forest soils" in this study and those from Ringenbach et al. (2022c). The derived necessity of future sensitivity analysis or even Monte-Carlo approaches would include the uncertainties raised by changing soil moisture.

## 5 Conclusions

In this work, we present a comprehensive experimental rockfall campaign comprising two rock shapes, three mass classes up to 3200 kg, and two states of the forest – with and without deadwood. The first key message relates to the unforeseen, inverse rock-shape effect compared with that in undisturbed open land: platy-shaped rocks are more affected by tree impacts, leading to a tilting away from their preferred wheel-like descending motion towards their flat side. This sliding leads to higher friction and shorter mean run-out distances than observed with cubic-shaped rocks. The incorporation of rock shape into rockfall hazard assessments remains inevitable, though with an altered premise for forested slopes. The treatment of platy-shaped rocks moving through densely forested slopes now becomes questionable. Nonetheless, in this experiment, the highest observed velocity, the longest jump, and the greatest run-out distance resulted from a platy-shaped rock that did not impact any trees. Forests suppress the lateral spreading of platy-shaped rocks only under the assumption of effective tree impacts. If no trees are hit or the rock is so large that the standing forest hardly influences its kinematics, lateral spread emerges again. A precise rockfall hazard assessment in forests requires, above all, a meticulous evaluation considering not only tree density but equally the locations of individual trees.

The second key message is that the observed protective effect of natural deadwood clusters is up to 1.7 times greater than the mean breakage work reported under laboratory conditions. This means it is reasonable to adopt the previously reported hazard reduction potential for larger rock masses. However, as the absorption coefficient of 415 kJ m$^{-2}$ indicates, for prevailing DBH distributions of mountain forests, the protection capacity of single logs becomes unessential when dealing with high-energetic

rocks. Additionally, the presented data corroborates values from the existing literature for the energy absorption of living spruce and beech trees, and it expands the scarce data set on rock–tree impacts.

Finally, the presented data set is currently the most comprehensive data archive available for single-block rockfall in forests and can serve the geohazard community as a calibration benchmark for various processes relating to rockfall kinematics in forests. A large number of reported impacts and the extremely detailed trajectory information make it possible to test existing or newly developed impact models and to further scrutinize rock–obstacle impact behavior.

In conclusion, rock shape still matters, with platy- and cubic-shaped rocks behaving differently on unobstructed and forested slopes. Accurate rockfall hazard assessments demand the incorporation of horizontal and vertical forest structures as accurately as possible in order to obtain realistic deposition patterns. Further interdisciplinary research is needed to enlarge the energy ranges, deal with the temporal evolution of the protective deadwood capacity, and increase the detail and input capacities of standard numerical models.

*Data availability.* The complete data set (deposition points, video footage, parameters of interest from every reconstructed trajectory, lidar point clouds) will be made publicly available in an EnviDat repository upon publication (Ringenbach et al., 2022b)

*Sample availability.* a sample of the video footage of a platy EOTA$_{221}$ rock under cleared conditions is (temporarily) online available under https://www.youtube.com/watch?v=rsgosdGe2Rg

*Author contributions.* AR and AC conceived the experiments. After forest inventories (AR) and optical UAV missions (planned and deployed by YB, AS and AR), the experiments were carried out by all authors of the corresponding data repository (Ringenbach et al., 2022b)). The second and third lidar UAV flights were conducted by AS and YB. AR edited the complete footage and reconstructed the rockfall trajectories. Based on the rockfall simulation (AR) and the data analysis (AR, AC), AR and AC wrote the manuscript, which was discussed and improved by all authors.

*Competing interests.* The authors declare that they have no conflict of interest.

*Acknowledgements.* We acknowledge partial funding by the National Research Program "Sustainable Economy: resource-friendly, future-oriented, innovative" (NRP 73) from the Swiss National Science Foundation (grant number: 407340_172415). Many thanks to the municipality of Schiers, for allowing us to use the forest, as well as to the Office for Forests and Natural Hazards (AWN), Region 1, for accompanying us. Further thanks to all the helping hands during fieldwork (which are also acknowledged as authorship of the data repository): Guillaume Meyrat, Miguel Sanchez, Gregor Ortner, Alex Bast, Mario Guetg, Gregor Schmucki, Julian Bleiker, Natalie Brožová, Thomas

Planzer, Michi Kuenz, Jessica Munch and Sophia Völk. We thank Matthias Paintner, SacaleVision, for the unique photo documentation, Philip Mayer, ETHZ, for the latest StoneNode hard- and firmware updates, Marco Collet, Silvio Burger, and Michael Hohl for enabling a waterproof StoneNode fixation and the remounting of their batteries. Geobrugg AG for additional acceleration sensors. Hansueli Rhyner for the rope safety devices and the thereof always accident-free camera assembly. LogBau Group For the crane-truck transport work and the considerable help with the large wheel loaders and the precision work carried out with them. HeliAir, for the rock transportation and

deadwood installations, and ROTEX Helicopter AG for the removal of the logs. Christian Ginzler, for organizing the first Aeroscout lidar flights, which thankfully reacted on the shortest notice, Mauro Marty for the help with the point cloud processing. Finally, Melissa Dawes for proofreading.

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
