# Peer review of "Shape still matters: Rock interactions with trees and deadwood in a naturally disturbed forest uncover a new facet of rock shape dependency"

_Earth Surface Dynamics, 2022_

## Author Comment (AC1)

**Author's response to the Interactive comment of Referee #1 on the manuscript "Shape still matters – rockfall experiments with deadwood reveal a new facet of rock shape relevance"**

Dear Louise M. Vick,

We thank you for the positive conception of our experimental rockfall campaign within a deadwood-containing forest. We highly value your opinion, that the presented findings are an important step in the research of natural hazards, both in terms of process understanding and rockfall simulation accuracy.

We appreciate the recognition of appropriately presented and discussed data. We understand your main criticism relates to our introduction and methods, which we will address in a revised version of the manuscript. We will mindfully expand the introduction and include the missing sections.

Regarding the title, we can understand the criticism that it does not fully capture the scope of the study. Therefore, we will consider a slight adaptation or extension to better reflect the study's complete range.

However, we disagree with the notion that rock shape is a small part of the experiment and results. Although initially intended as a side story, the observed inverse shape effect within forests compared to open land is a novel and unexpected finding, with significant implications for rockfall hazard assessments. While the rock shape may have been underrepresented in the introduction, we expect that this discovery will have a considerable impact on future requirements for rockfall hazard assessments, highlighting the often neglected importance of rock shape.

Further, we will amend the experimental design section. Here we address your particular questions:
- The $EOTA_{111}$ rock shape is the original test block geometry used for certification tests of flexible rockfall protection systems (e.g. Volkwein et al., 2019). This rock has three symmetrical axes, represented by the index 1 per axis. The $EOTA_{221}$ rock shape is the platy version of the $EOTA_{111}$ with two longer axes and one shorter axis, as described (l. 74-75) and illustrated (Fig. 4) in the original manuscript.
- "tree mass classes" was a typo. We used **th**ree different mass classes, consisting of roughly 200 kg, 800 kg and ≥2600 kg rocks of both shapes. The ≥2600 kg class is a class, as it consists of 2600 kg and 3200 kg rocks. But also the other mass classes are correctly entitled as classes, as slight mass differences between the rock shapes were noticed.
- In total, we released 106 artificial concrete rocks with masses > 200 kg into the forest. Therefore we cast different rocks in 8 different molds ($EOTA_{111,200 \text{ kg}}$, $EOTA_{111,800 \text{ kg}}$, $EOTA_{111,2600 \text{ kg}}$, $EOTA_{111,3200 \text{ kg}}$, $EOTA_{221,200 \text{ kg}}$, $EOTA_{221,800 \text{ kg}}$, $EOTA_{221,2600 \text{ kg}}$, $EOTA_{221,3200 \text{ kg}}$). Here comes the advantage of this approach: It is irrelevant how many different samples we used per mold, as they were standardized shapes, and their mass was within minor variances. But
- The number of released rocks per mold and the state of the forest are indicated in the table below and will be (slightly adapted) also in the next version of the manuscript:

Table 1: Overview of the conducted experimental runs. bold: reconstructed trajectories.

| | DW org | CLR | DW installed |
|---|---|---|---|
| $EOTA_{111,200 kg}$ | 12 | 8 | |
| **$EOTA_{111,800 kg}$** | **10** | **10** | |
| **$EOTA_{111,2600 kg}$** | | **2** | **2** |
| **$EOTA_{111,3200 kg}$** | | **2** | **2** |
| $EOTA_{221,200 kg}$ | 11 | 12 | |
| **$EOTA_{221,800 kg}$** | **10** | **13** | |
| **$EOTA_{221,2600 kg}$** | | **3** | **3** |
| **$EOTA_{221,3200 kg}$** | | **3** | **3** |

- The number of 13 trajectories mentioned (l. 190) arose from all $EOTA_{221,800 kg}$ runs within the CLR forest state. This is the one, with the most single experimental runs per shape/mass and forest state.

Additionally, the response to the minor comments are given below:

***Can the authors define early on what is meant by deadwood in this particular case- trees which have died but remain fully standing, or trees which are broken?***

As we are talking of wind**thrown** forest, we had lying deadwood in mind (as depicted in Fig. 1.g). Indeed, we did not explicitly mention the state of the deadwood, and we will address this in a revised version of the manuscript.

***Figure 1a. Why are the green points not displayed according to size like the blue?***

Although we already included the different DBH classes within the legend, the icon size was - by mistake - not varied between them. We will adapt this shortcoming in a new manuscript version.

***L182: How is this consideration threshold derived?***

The threshold itself is set based on field observations. The rocks impacting deadwood logs with a diameter < 20 cm did a) easily break or b) roll over them. The breakages are plausible, as rather low rock velocities of 11 ms$^{-1}$ fulfill the stated breakage condition, assuming the smallest rock mass (200 kg). This velocity is even reduced, to 6 ms$^{-1}$ or roughly 3 ms$^{-1}$ for the larger rock masses 800 kg and 2600 kg. Also, the overcoming of deadwood logs with a diameter < 20 cm seems likely, as the center of mass is even for the smallest rock ($EOTA_{111, 200 kg}$) 25 cm above the ground.

***L183-4: What is a root plate and what is the purpose of this step?***

Root plates are discs of soil and ground material, formed by the root system of an overturned tree. During some experimental runs, we observed, that deadwood breakage happened, but never the less, the rock was stopped behind a root plate. As it is not common in forestry practice to clear the root plates we left them not only during the CLR-experimental campaign but also in their simulations.

***L191: How were these parameters calibrated? What field data went into this?***

The calibration was based on a manual assessment. Of the mentioned parameters (l. 190). We aimed for a reproduction of the deposition pattern, both, the longitudinal (maximum run-out length across the river), and the lateral (several rock depositions in the adjacent terrain chamber). Based on the reconstructed trajectories we knew that maximal velocities of >30 ms$^{-1}$ occurred for those rocks, with barely any tree impacts. This allowed us to fine tune already known forests soil parameters from other studies (e.g. Ringenbach et al., 2022)

**L201: Does MDH mean resting elevation of the block? Unclear**

As stated in l. 161 – 164 we analyze the median deposition heights. However, we neglected to introduce the abbreviation in that section, which we will rectify in a future manuscript.

**L311: How can it be a pattern and also not statistically significant?**

A pattern in itself could be statistically insignificant. However, in this case, we based our statement of insignificance on the overlapping boxplot notches of the 0.256 and 0.302 VWC. We recognize that this analysis is incomplete, and we will provide a more comprehensive statistical analysis in the revised manuscript.

**F9: What are the white lines crossing the slope?**

The white lines depicted in the figure represent the deadwood logs, which are included within the RAMMS::Rockfall model. While the current caption lacks detail, we will address this issue and provide a more comprehensive caption in the revised manuscript.

**L482: This is untrue. See for example https://doi.org/10.5194/nhess-19-1105-2019**

Thank you for pointing out this study. We will adapt the corresponding sentence, amend the whole paragraph, and shift parts of it into the introduction.

---

## Author Comment (AC2)

**Author's response to the Interactive comment of Referee #2 on the manuscript "Shape still matters – rockfall experiments with deadwood reveal a new facet of rock shape relevance"**

Dear Christine Moos,

Thank you for your positive feedback regarding our "very impressive and valuable" results, which you believe are "definitely worth publishing." We are delighted that you have an overall positive perception of our work.

The primary critique of our manuscript is the lack of a coherent narrative or "red thread." While we appreciate your recognition of our results' value and importance, reducing the overall content is not a feasible solution. We acknowledge that the manuscript encompasses four main topics, including the impact of rock shape on the deposition pattern, the correlation between rock and deadwood impacts, the limited influence of soil moisture, and the simulation of high-energy scenarios involving deadwood. Although the current scientific climate may call for splitting such content into several micro-publications, we question whether such an approach would be in the best interest of the genuine reader.

Given that our study comprises a comprehensive and integrated experimental design, we intend to present it as such.

Nevertheless, we are committed to finding a solution to this problem and will attempt to highlight the coherent narrative within the existing manuscript better. To this end, we have focused on the two most significant trajectories out of the 106 possible options, which we have used to describe the rolling and sliding trajectory reconstruction methods and to explain the deposition pattern shown in Figure 2b. We have referenced this information again in Figure 6f, discussing the maximum observed energy absorbed by a standing tree. Following our evaluation of the impact on standing trees, we analyzed the effects on the lying deadwood using the same methodology.

We will supplement the title in line with the proposed suggestion., while still prominently featuring the rock shape effect, as we previously indicated in response to Referee 1.

We appreciate your feedback and are dedicated to addressing all of the comments in the revised manuscript, which will be substantially improved as a result.
Specific comments

**Title: Why "still" matters?**

Lately, publications (Caviezel et al., 2021, Bourrier et al., 2021) mentioned the importance of rock shape on deposition patterns during open land rockfall experiments. With the "still "we refer to them, while the "new facet" points out the new, contradicting findings within the forests. However, we will also amend the title to include the other parts of the study.

**L1: Sentence "Rates of deadwood production have already increased" sounds very general and I am not sure, whether this fact applies to all regions and time periods. Please be more specific (regarding where and time period > in past few years, decades,…)**

We will add the geographical and the time context to that general sentence.

***L23&24: Consider replacing "ecological" by "ecosystem-based" or "nature-based" (or "green"), to use one of the most common terms in this field***

We will also include "nature-based", as this term is also precise. "green" is among our understanding too unspecific.

***L24: Consider replacing "accepted" by "recognized".***

We will amend the manuscript with recognized.

***L32: The references for the disturbances seem arbitrary. I suggest deleting them and eventually complement sentence with a reference that underpins your statement (that disturbances have been neglected in models)***

We agree that the sources could be deleted.

***L33: It is not obvious, why natural disturbances necessarily have to be integrated in numerical tools, but rather their effect on the protective effect of forest should be quantified (and numerical models can be a tool therefore). This is e.g. also what is done in part of the references (Fuhr et al., 2015, Costa et al., 2021). Please reformulate more clearly.***

We will change the manuscript and will underline the importance of incorporating the affected protective effect of forests, rather than the natural disturbances themselves.

***L74: Why "mass classes" and not "masses"? You report a single weight and not a class (except for the largest blocks)***

Already your question is the most important part of the answer: as we did not have enough repetitions of the rock masses >2600 kg, it was necessary, to combine the 2600 kg and 3200 kg rocks into a mass class (See the table provided in answer to referee 1). Additionally, the other mass classes were entitled as such, as slight mass differences between the rock shapes did occur. We will pronounce these two points more concise in a new manuscript version.

***L78: Not clear, why the deadwood was removed. Please reformulate more clearly.***

We amended the section and clarified, that the destructive potential of the largest rock mass class would have changed the appearance of standing forests too much. Therefore the comparison experimental campaign CLR with the cleared deadwood section would have been corrupted for all three mass classes (200 kg, 800 kg, >2600 kg). Only the DW campaign for > 2600 rocks was slightly biased with the chosen procedure, as the relevant deadwood logs were reinstalled.

**L98: Is it general knowledge that an Airbus H125 is a helicopter?**

We will refine the vehicle type

**L100: Where were the soil moisture sensor installed (one for enire slope? Several?) (only reported in Figure 1)**

There was a single data logger, at the position reported in Fig. 1. Due to a failure, the initially redundant measurements were only for several time steps available for one sensor per depth, which were further used within this study.

**L102: What do you mean by "according to their availability"?**

By "according to their availability", we mean that we did not install the same amount of sensors every time, as some of them were borrowed and not during every experimental day available.

**L106: I do not think that it is necessary to explain previously used trajectoryreconstruction methods, but to describe the method used in this study and explain why.**

The complete explanation of why we used the finally deployed method contains the shortcomings of the previously introduced methods. Without such a short literature review, the reader would feel lost. This introduction could be split into some sentences in the introduction, methods and discussion. However, a division would further prove the criticism of the missing thread, since there are only a few sentences in each of the three chapters, which would have too little to do with the rest of the paragraph.

**L143: How are the classes "soft", "hard",… defined?**

We defined hard impacts as such when some rock dust during/after the impacts were visible on the footage. We will add this information in a subsequent version of the manuscript.

**L201: Do you mean the mean run-out distance with "mean deposition height"? Or is it an elevation and if yes, why? The Expression is confusing.**

As described in the methods section (l. 161 – 164) we do not analyze the run-out distance but analyze the altitude above sea level of the deposited rocks. We did this due to the complex topography, meaning the general bend of the trajectories, the fluvial terraces, and the changing riverbeds between the experiments. These terrain features would not allow for comparing the mean run-out distance in an unbiased manner. Nevertheless, we agree that the so-far used expression *mean deposition height* is confusing and will rename it to mean deposition altitude.

***L203: Is the MDH reduction statistically significant? Did you do any statistical test? It could be interesting to see the actual distribution of run-out distances (with and without deadwood) and not only the mean values.***

Fig. 10. a and b) of the original manuscript shows the asked distribution of the deposition altitudes. Based on the non-overlapping notches of the boxplots, we can assume that deadwood reduced the mean deposition altitude significantly (95 % confidence interval) for $EOTA_{111, 200\,kg}$, and both shapes of $EOTA_{800\,kg}$. We will add the missing reference to this figure in the next version of the manuscript.

***Figure 5: Would be nice to see the boxplots for deadwood and cleared next to each other (per zone). I am not sure whether Fig. c) and d) are necessary – could be moved to the appendix to avoid an "overload" of plots.***

We see advantages in the current figure setup: Thanks to the current figure structure, it is possible to work with the same color code as in e) to intuitively georeference the data in a) and b). Referencing would barely be possible as the color would be used to differentiate between the DW and CLR states of the forest. However, as we used the same y-axis scale for both states, a comparison is already now possible.

Subplots c) and d) are further crucial for the (apparently) missing "red thread": they focus on the rock shape relevance and reveal that platy-shaped rocks feature lower rock velocities (95$^{th}$ percentile)

***Caption:***

***In-depth velocity comparison***

***…all screens span the entire width of the slope***

We do not see the point in question

***L224: Not necessary to mention that a statistical analysis indicated. Suggestion for reformulation: "The mean velocity increased by …".***

We will amend the section in question.

***L226f: Confusing sentence since you use to reasonings: "Consequently,.." and "due to" (what is now the reason for what?)***

We will clarify the confusing sentence by splitting it into two sentences.

***L225: Here again: Did you perform a statistical testing? Are the differences significant?***

The significance was not tested in the state of the original manuscript. Nevertheless, all the differences are statistically significant. We will amend this section and provide the additional information in a new manuscript version.

*L233f: This information belongs to the method part.*

We used to see these two sentences more as a link within the text for Figure 5a-d, which is definitely a result. But we agree that the reference and the introduction of the evaluation screens (S1-10, Fig. 5e) have a methodological character. Therefore we will remove this section from the results chapter and introduce them within the methods chapter. We would not introduce the figure differently. So adding these sentences to the methods section would result in redundancy.

*L248f: Again, you report here methodological details, which should not be part of the Results section.*

We agree that straightforward, pre-experimentally defined methods have to be reported in the corresponding section. Here we argue, that it was possible solely after the experiments, to define the exact procedure (=fitting upper and lower line), after consideration of the first results.

Based on this view, we moved equation (1) to the methods section. It is the basic concept we thought about already before knowing our results, while comparing the data from Dorren et al, (2005), Ammann (2006) and Kalberer et al. (2007). See also your comment L255. However, it would not be clear to the reader why we apply a two-way process for standing trees while we plot only one line for deadwood without considering the results. We were forced to adapt the applied methods during the data analysis, based on the wood conditions (living vs. deadwood). Therefore, it makes sense to explain this issue in the results section, based on the knowledge the reader has at this stage.

*L255: The equation for the fitted absorption relationship should be moved the Method sections. Only report results here.*

We agree, under consideration of the answer to the above comment L248f.

*Figure 6: Mention difference between a) and b) (Fagus / Picea) in caption.*

We will clarify the missing species also in the caption and not only (as in the original manuscript) graphically.

*L298: You begin the paragraph with "experiments were solely held during dry conditions". Later on, you write "while the three experiments in […] were carried out under rather moist conditions […]". What is now the case?*

Although we thought, that within context, the meaning should be clear, we agree, that additional clarity will arise, if we amend this paragraph as mentioned in the following: "experiments were solely held during dry **weather** conditions" and "while the three experiments in […] were carried out under rather moist **soil** conditions"

*L300: Are the exact times of the measurements necessary? I think most important are the measured ranges of soil moisture content.*

The exact times are indeed not of greater relevance. They could have been used to check for infiltration rates of the rainwater. As we do not discuss this issue further, we will delete the detailed information in the future manuscript.

*L314: The correlation between velocity change and VWC seems rather weak, and I am asking myself whether it makes sense to fit a function to the relationship? How good is the fit?*

Obviously, the fit is not very strong. We pointed that out, by plotting also the 95 % prediction interval, which is > 10 m s$^{-1}$. Although without such a calculated and plotted fit function, the weak goodness of fit could not be reported.

*L326: Here again: "deposition heights" is confusing: Do you mean the elevation of the depositions?*

See answer for comment L201.

*L356: Here it comes – but although this new insight on the relevance of rock shape is very interesting, it is only one aspect among many in the study objectives and results and, thus, too dominating in the title in my view.*

We structured our discussion first in a general section, followed by the most important and interesting results. So we are of the opinion, this is the most prominent position to expect "it". As mentioned, we will adapt or extend the title without removing the shape relevance.

*L388: The (longer-term) effect of the decay of the deadwood on its protective capacity is rather relevant for protective management. Would be good if you could elaborate on that.*

Due to the supposed lack of a red thread, we should not open different side stories. This study focused mainly on fresh deadwood (as mentioned in your comment L503). We already stated in the original manuscript the reduced shear force capacity of a deadwood log over time. Further, we identified further research about the temporal evolution of the protective deadwood capacity (L515). At this position, we further add a source which discusses this matter in more detail.

*L393: You might refer here to Toe et al., 2016 (https://link.springer.com/article/10.1007/s10346-017-0799-6) , who conducted a sensitivity analysis on the parameters influencing the energy reduction.*

Thank you for pointing out that possible cross reference.

*L408: The integration into what?*

We will clarify the mentioned sentence in the new manuscript. However, we meant the integration of living and dead trees into numerical rockfall models.

*L450: Sentence unclear. Please reformulate.*

We will clarify the sentence and delete the unnecessary part of it.

*L457: The sentence "Such retrieval of kinetic energy was not observed for platy-shaped rocks, because of the greater protection of the standing forest" is not clear to me: is the protection of standing trees only greater for platy-shaped rocks?*

We will clarify the sentence. Thus, it is true that the intact forest provides substantial protection against platy rocks, as demonstrated by our experiment. We observed that cubic blocks resumed rolling after being obstructed by trees, whereas platy blocks often remained in contact with the ground and stopped completely. Therefore we keep repeating: shape matters.

*L477: As mentioned before, the soil moisture part is not very well embedded in the whole "story" of the article (in particular in the Introduction and Results). Here you raise some interesting aspects, but the link to the protective effect of the forest could be enhanced (e.g., soil moisture probably tends to be higher in forests compared to open land and, thus, this would even increase indirectly the protective effect of forests).*

We will strive for a better embedment of the soil moisture within the text. We plan to achieve this within the introduction, not by adding questionable conclusions. Because your comment on L314, where you even question to fit a function at all, and this additionally proposed discussion point, that based on higher soil moisture forests, higher protective effects compared to open land could be expected, are strongly contradicting.

*L503: …protective effect of natural fresh deadwood…*

We will add the state of the deadwood decay in this sentence.

*L515: Here you briefly mention the temporal evolution of the protective deadwood capacity. As stated before, I think you should discuss this more thoroughly.*

See the answer to the comment L388: We see the focus on fresh deadwood and will not focus too strongly on its temporal evolution.